

# Diatoms as paleoproductivity proxy in the NW Iberian coastal upwelling system (NE Atlantic)

Diana Zúñiga[1,2], Celia Santos[3,4], María Froján[2], Emilia Salgueiro[3,5], Marta M. Rufino[3,5], Francisco De la Granda[6]. Francisco G. Figueiras[2], Carmen G. Castro[2], Fátima Abrantes[3,5]

[1] University of Vigo, Applied Physics Department, Campus Lagoas Marcosende, E-36310, Vigo, Spain
[2] Consejo Superior de Investigaciones Científicas (CSIC), Instituto de Investigaciones Marinas (IIM), E-36208, Vigo, Spain
[3] Instituto Português do Mar e da Atmosfera (IPMA), Div. Geologia e Georecursos Marinhos, 1749-077, Lisbon, Portugal
[4] MARUM, Center for Marine Environmental Sciences, Leobener Strasse, 28359, Bremen, Germany.
[5] CCMAR - Centre of Marine Sciences. Universidade do Algarve, Campus de Gambelas, 8005-139 Faro
[6] Bundesamt für Seeschifffahrt und Hydrographie, Bernhard-Nocht-Str. 78, 20359, Hamburg, Germany

*Correspondence to*: Diana Zúñiga (diana.zuniga@uvigo.es; imissons@gmail.com)

**Abstract.** The objective of the current work is to better understand how diatoms species determine primary production signal in exported and buried particles. We evaluated how the diatom's abundance and assemblage composition is transferred from the photic zone into the seafloor sediments. A combined analysis of water column, sediment trap and surface sediment samples recovered in the NW Iberian coastal upwelling system was used.

Diatom fluxes ($2.2 \pm 5.6 \ 10^6$ # valves $m^{-2} \ d^{-1}$) represented the majority of the siliceous microorganisms sinking out from the photic zone and showed strong seasonal variability. During downwelling seasons, diatoms export signal was strongly affected by resuspension of bottom sediments and intense Minho and Douro riverine inputs, with benthic and freshwater diatoms (17 - 24%) becoming relevant in the sediment trap assemblage. Nevertheless, during upwelling productive seasons, the diatoms exported out from surface layer reflected water column diatom assemblage. They were principally represented by *Chaetoceros* spp. (mean 46 ± 25%) and *Leptocylindru*s spp. (mean 20 ± 22%) resting spores, demonstrating that both groups are a good sedimentary imprint during highly productive periods. Moreover, our data showed that the sink of *Chaetoceros* spp. resting spores dominated under persistent upwelling winds, high irradiance levels and cold and nutrient-rich waters. Otherwise, *Leptocylindrus* spp. spore fluxes were favoured when northerly winds relax, and surface waters warming promotes water column stratification. Further, this finding will provide a proxy of persistent vs. intermittent upwelling conditions, which is of particular relevance in palaeoceanography.

**Keywords:** diatoms; coastal upwelling; organic carbon; biogenic silica; sediment trap; NW Iberian;



## 1 Introduction

The ocean plays a critical role in the global carbon cycle as a vast reservoir that takes up a substantial portion of the anthropogenically-released carbon from the atmosphere (LeQuéré et al., 2009). A key aspect to understand the ocean carbon cycle includes the fixation of atmospheric $CO_2$ by diatoms (Falkowski et al., 1998; Smetacek, 1999; Boyd and Trull, 2007).

These microorganisms, which are the most important global producers respond to environmental characteristics in the upper water column. In addition, they play a key role as sinkers of particulate organic carbon and biogenic silica from the surface productive layer to the sediment record (Sancetta, 1989; Romero and Armand, 2010; Tréguer and De La Rocha, 2013). All these points underpin the importance and the effectiveness of diatom species as productivity indicators in Earth´s climate system studies. Nevertheless, primary production reconstructions suffer from the uncertainty about how diatoms respond to

particular environmental conditions, and how particular diatom species transfer primary production signal via exported and buried particles. Therefore, to better understand and quantify past productivity conditions from sediment cores, regional calibrations are required.

Coastal upwelling systems as sites of major primary production, with a marked seasonality are thus ideal for these types of studies (Walsh, 1991; Falkowski et al., 1998; Capone and Hutchins, 2013). Consequently, many works have been focused on

the study of diatom fluxes in highly productive coastal regions with the aim to better understand the diatom signal at longer time scales (Sancetta, 1995; Lange et al., 1998; Romero et al., 2002; Abrantes et al., 2002; Venrick et al., 2003; Onodera et al., 2005).

In the NW Iberian continental margin, seasonal upwelling favouring winds generate high primary production rates through modulation of the microplankton community structure (Figueiras and Pazos, 1991; Nogueira and Figueiras, 2005; Espinoza-

20 González et al., 2012). Although there are no previous studies using long-term sediment traps for the NW Iberian margin, several works have assessed the diatom species ecology - environmental conditions relationship by comparing the recent sediment record to the hydrographic conditions (Abrantes, 1988; Bao et al., 1997; Abrantes and Moita, 1999; Gil et al., 2007; Bernárdez et al., 2008; Abrantes et al., 2011). Those authors concluded that the spatial distribution of the sedimentary diatom abundance and assemblages´ composition reflects the hydrographic upwelling patterns and primary production

trends, with *Chaetoceros* resting spores appearing as a good tracer of the upwelling regime.

The aim of this study is to evaluate the use of marine diatoms as a paleoproductivity proxy for the NW Iberian coastal upwelling system. This work presents the first diatom assemblage analysis that combines samples from the water column, sediment trap and surface sediment samples to understand the mechanisms regulating the export of specific diatoms species from the photic zone into the seafloor sediments.

## 30 2 Regional setting

Our study site (RAIA station) is located in the NW Iberian continental shelf off Cape Silleiro (42º 05´ N; 8º 56´ W at 75 m water depth, Fig. 1). During spring – summer, the NW Iberian coast is characterized by prevailing northerly winds, that



favour upwelling of cold and nutrients rich subsurface Eastern North Atlantic Central Water (ENACW) on the shelf and into the Rías, resulting in a primary production increase in the area (Fraga, 1981; Fiuza, 1984; Tenore et al., 1995: Figueiras et al., 2002). In contrast, south-westerly winds favour coastal downwelling during autumn-winter. From October to January the region is generally affected by the northward advection of warm, saline and nutrient-poor southerly waters by the Iberian Poleward Current (IPC) (Haynes and Barton, 1990, Relvas et al., 2007). Later on, usually between February–March, a decrease of temperature associated with winter cooling leads to a well homogenized mixed layer of cold and nutrient rich waters (Álvarez–Salgado et al., 2003; Castro et al., 2006). In addition, during downwelling seasons, the occurrence of south-westerly winds in this high-energy dynamic margin can generate moderate to extreme storms with wave heights > 6 m, which have been simulated to produce high sediment remobilization (> 50 %) (Vitorino et al., 2002; Jouanneau et a., 2002; Oberle et al., 2014). During these hydrodynamic periods, this site is also highly influenced by the Minho and Douro Rivers discharge, which provide important sources of terrestrial sediments to the inner shelf (Dias et al., 2002).

## 3 Material and methods

### 3.1 External forcing

Irradiance data was obtained from Cies meteorological station (IR; 42° 13′N, 8° 54′W; 25 m height) supported by Galician government and accessed at MeteoGalicia website (www2.meteogalicia.es) (Fig. 1).

Daily Ekman transport ($-Q_x$), an estimate of the volume of upwelled water per kilometre of coast was calculated according to Bakun´s (1973) method:

$$-Q_x = -((\rho_a \, CD \, |V|) / (f \, \rho_{sw})) \, V_y$$

where $\rho_a$ is the density of the air (1.22 kg m$^{-3}$) at 15 º C, CD is an empirical dimensionless drag coefficient (1.4 10$^{-3}$), f is the Coriolis parameter (9.76 10$^{-5}$) at 42 ºN, $\rho_{sw}$ is the seawater density (1025 kg m$^{-3}$) and |V| and $V_y$ are the average daily module and northerly component of the geostrophic winds centred at 42º N, 10º W, respectively. Positive values show the predominance of northerly winds that induces upwelling on the shelf and negative values indicate the presence of downwelling. Minho and Douro River discharges were obtained from https://github.com/PabloOtero/uptdate_rivers (Otero et al., 2010). Wave data were based on WANA hindcast reanalysis of 3027034 (WANA$_S$: off Silleiro: 42º 15'N; 9º W) and 1044067 (WANA$_G$: off A Guardia: 41º 45′N; 9º W) points, and supplied by Puertos del Estado (Fig. 1).

### 3.2 Water column

RAIA station (75 m water depth) was visited monthly on board "R/V *Mytilus*" from March 2009 to June 2012 except during the period December 2009-June 2010. Characterization of the water column was conducted by i) CTD-SBE25 profiling and



ii) collection of discrete water column samples using a rosette sampler with 10-LPVC Niskin bottles for inorganic nutrients, chlorophyll $a$ (Chl $a$) analysis and diatoms counting and species identification.

Water column stability (0-35 m) was analysed by using Brunt Väisälä frequency parameter, $N^2 = [g / z] \ln (p_z/p_0)$ where g is the local acceleration of gravity, z is the water depth and $p_z$ and $p_0$ the bottom and surface density, respectively.

Nutrients concentration was determined by segmented flow analysis with Alpkem autoanalysers (Hansen and Grasshoff, 1983). The analytical errors were ± 0.05 μmol kg$^{-1}$ for nitrate and silicate and ± 0.01 μmol kg$^{-1}$ for phosphate. Final Chl $a$ concentrations were determined by pigment extract fluorescence using a Turner Designs fluorometer calibrated with pure Chl $a$ (Sigma) (please see details in Zúñiga et al., 2016).

For diatoms counting and identification a seawater volume of 100 ml collected at 5 m water depth was preserved with
Lugol´s iodine until microcoscopic observation. Depending on the water column Chl $a$ concentration volumes between 10 to 50 mL were deposited in composite sedimentation chambers and observed through an inverted microscope. The microorganisms were counted and identified to the species level, whenever possible, using the Utermöhl sedimentation method (Utermöhl, 1931, 1958). For this study only water column diatom species that appeared in more than one sample with a percentage higher than 2 % of the total abundance were considered for further analysis primary production.

**3.3 Sediment trap**

RAIA station was monitored with an automated cylindric-conical Technicap PPS 4/3 sediment trap (height/diameter ratio of 1.7 and a collecting area of 0.05 m$^2$) from March 2009 to June 2012 (Zúñiga et al., 2016). The trap was deployed at 35 m water depth over sampling intervals between 4-12 days. Examination of CTD pressure data mounted 2 m below the sediment trap showed that the mooring line tilting was less than 5º during 70 % of the time it was deployed. Only in exceptional cases
highly hydrodynamic events lead to velocities higher than 25 cm s$^{-1}$ and mooring tilts between 15-20º. Therefore, we can assume that the trap was not affected by hydrodynamical biases. Sampling strategy and samples processing details are explained at length in Zúñiga et al. (2016).

Total mass flux was gravimetrically determined. Biogenic silica content was analysed following Mortlock and Froelich, (1989). The samples were treated with 2M $Na_2CO_3$ for 5 h at 85 ºC to extract the silica and then measured as dissolved silica
by colorimetric reaction. Biogenic opal was converted from Si concentration after multiplying it by a factor of 2.4.

Sample preparation for diatom abundance and assemblage assessment was adapted from Abrantes et al. (2005). Depending on the recovered material 1/5 or 2/5 splits of the original samples were used, after rinsing $HgCl_2$ by repeated settling in distilled water. Subsequently, organic matter and carbonates were removed by the addition of $H_2O_2$ (30 %) and HCl (10 %), respectively. Permanent slides were prepared using the evaporation-tray method of Battarbee, (1973) and mounted with
Norland optical adhesive (NOA61). Diatoms counting and species identification was performed at 1000 X (10 x eyepieces and 100 x objectives), using a Nikon Eclipse E100 microscope equipped with Differential Interference Contrast (DIC). 100 randomly selected fields of view were counted in 3 replicate slides (Abrantes et al., 2005). Diatom flux was calculated as





F= ((N)(A/a) (V)(S)(X))/D

where the flux F is expressed as number of valves $m^{-2}$ $d^{-1}$, N is the number of valves counted in 100 randomly selected fields of view, (a) represents the counted fraction of the total tray area (A), V is the dilution volume, S is the split fraction, X is the conversion factor from the collecting area to 1 $m^2$, and D is the sampling interval in days for each sample.

5    Relative abundance of diatom taxa was determined following the counting procedures from Schrader and Gersonde, (1978) and Abrantes, (1988). For each sample, ca. 300 individuals were identified to the lowest taxonomic possible level, and raw counts were converted to percentage abundance. In samples containing low diatom abundances, the number of individuals identified was 100 – 200 (Fatela and Taborda, 2002). For this study only sediment trap diatom species that appeared in more than one sample with a percentage higher than 2% of the total abundance were considered for further analysis.

## 3.4 Surface sediments

Surface sediments (0 - 1 cm depth interval) were recovered from GeoB 11002-1 (42º 10´N N, 8º 58´ W; 111 m) and GeoB 11003-2 (42º 10´N, 9º 1´ W; 129 m) stations located near the RAIA position (Fig. 1). Samples were collected in August 2006, using a giant box corer during the GALIOMAR expedition (P342) on board of the R/V *Poseidon*. Sample cleaning and slides preparation of the samples was carried out following the methodology of Abrantes et al. (2005). Counting and
identification procedures were the same as for sediment traps samples.

## 3.5 Statistical data analysis

Relationships between environmental variables and diatoms species from the sediment trap record were evaluated with Pearson correlation coefficients (Table 1 and 2).

The relationships between the main groups of diatoms (freshwater diatoms, benthic diatoms, *Chaetoceros* resting spores,
*Leptocylindrus* resting spores and *Paralia sulcata*) and the environmental variables were analysed using the ordination technique Canonical Correspondence Analysis (vegan package, R-project (ter Braak, 1986; Oksanen et al., 2015). Environmental data used for this analysis resulted from water column data interpolation by considering sediment trap sample recovering intervals. Resulting data were subsequently integrated to 35 m where the sediment trap was moored. The multicollinearity of environmental variables was previously tested by Pearson correlations (Dormann et al., 2013) and
checked after modelling using variance inflation factors (VIFs) applied to the CCA. Nine environmental variables were thus initially included in the ordination: irradiance, temperature, Brunt Väisälä frequency parameter ($N^2$), Chl *a*, $NO_3$, $Si(OH)_4$, upwelling index (UI), Minho River flow, A Guarda wave height (Hs). Significant environmental variables were identified via a stepwise procedure, using permutation tests (999 permutations). After the selection of the significant variables, the model was tested a second time through a Monte Carlo global permutation test (999 permutations) to assess the significance
of ordination axes.




The results of CCA were presented as ordination bi-plot diagram containing the explanatory variables plotted as arrows along with points for samples (dates) and species (main groups of diatoms. Using these diagrams, we were able to identify the relationships between species, between samples, and relationships of samples and species to environmental variables.

# 4 Results

## 4.1 Environmental conditions and water column characteristics

From October to April-May, the NW Iberian margin was generally characterized by the prevalence of low irradiance levels and south-westerly winds as shown by the negative $-Q_x$ values (Fig. 2a and 2b). This downwelling season was also accompanied of strong SW storms promoting wave heights frequently higher than 4 m and intense Minho and Douro River discharges (Fig. 2c and 2d). Hydrographically, in a first phase we can distinguish the presence of the IPC (October-January) characterized by anomalously warm water (15-17 ºC) with relatively low nutrient and Chl a ($< 4$ mg m$^{-3}$) content (Fig. 3). Later on, we differentiate the mixing period with temperatures around 14 ºC and higher nutrient levels due to winter cooling. Furthermore, river runoff promoted water column stability through the formation of low salinity water lens at sea surface with significant loads of silicate-rich particulate matter (Fig. 2c, 3a, 3b and Table 1). During downwelling periods, diatom abundances were low ($\sim$14 cel mL$^{-1}$) (Fig. 3c). Small centric cells accounted for the largest shares (37–84 %) and only exceptionally *Navicula* spp. and *Paralia sulcata* become relevant (Fig. 4b and 4d). From May to October, when irradiance levels remained high, a series of upwelling relaxation events promoted the upwelling of cold ($< 14$ºC) and nutrient rich ENACW on the continental shelf, leading to the development of Chl *a* maxima (Fig. 3). During the highly productive upwelling seasons, diatom abundances achieved maximum levels (up to 7629 cel mL$^{-1}$) (Fig. 3c), and the predominant genera were *Chaetoceros* and *Leptocylindrus* spp. (Fig. 4f). Other species frequently associated with upwelling favourable conditions (e.g. *Asterionellopsis glacialis*, *Detonula pumila* or *Guinardia delicatula*), appeared in the upper water column, sporadically and with lower abundances (Fig. 4g and 4h).

## 4.2 Sinking particulate material time series

Time series of the siliceous organism fluxes calculated from microscopic counting matched biogenic silica fluxes registered by the trap, which contribution to the total material ranged from 2 % to 10 % (Fig. 5a, 5b and 5c). The contribution of diatoms to total siliceous microorganisms was high during all study years (Fig. 5c and 5e). Only during the 2012 upwelling season, silicoflagellates become relevant, achieving more than 7 % of the total siliceous organisms (Fig. 5d). Maximum total diatom fluxes (up to up to 22.6 10$^6$ # m$^{-2}$ d$^{-1}$) were registered under downwelling conditions when benthic and freshwater



diatoms fluxes became relevant, with contributions to the total diatom fluxes up to 24 % and 17 %, respectively (Fig. 5e and 6).

Regarding the fluxes of marine diatom assemblage, during downwelling periods, small centric diatoms and additionally *Paralia sulcata* significantly contributed to the total marine diatom fluxes with percentages achieving maximum values of 11 % and 54 %, respectively (Fig. 7a, 7b, 7c, 7d). On the contrary, under upwelling favourable conditions, the marine diatom assemblage found in the trap were mainly composed by *Chaetoceros* and *Leptocylindrus* spp. resting spores with contributions to total marine diatom fluxes fluctuating around mean seasonal values of 46 % and 20 %, respectively (Fig. 7a, 7b and Table 3).

## 4.3. Surface sediment samples

Diatom abundances in GeoB 11002-1 top sediment sample was significantly higher than in the offshore GeoB 11003-2 station with values of 142 x $10^4$ # valves $gr^{-1}$ and 65 x $10^4$ # valves $gr^{-1}$ respectively (Table 3). Diatom assemblages were similar at both sampling sites, with predominance of *Chaetoceros* spp. (33-39%) and *Leptocylindrus* spp. (37-39 %) resting spores and *Paralia sulcata* (10-17 %) (Table 3). Otherwise, benthic and freshwater diatoms have a significantly lower contribution (< 4 % and 1 %, respectively) at both stations.

## 4.4 Relationships between sediment trap main diatom groups and environmental variables

Canonical correspondence analysis (CCA) stepwise procedure identified five significant variables (Minho River flow (Minho river), temperature (Temp), Chlorophyll *a* (Chl *a*), $NO_3$ and $Si(OH)_4$) for the abundance of the main diatom groups (p-value < 0.05) (Fig. 8). The first two canonical axes explained 48.7 % and 40.4 %, i.e. 89 % of the modelled inertia and consequently only those two axes were considered. The CCA model with the five variables explained 46% of the total inertia. The first canonical axis showed a positive gradient with Temp and Chl *a* opposite to Minho River discharge. Freshwater (FW) diatoms, benthic diatoms and *Paralia sulcata* (Parsul) were negatively positioned in the first canonical axis, indicating thus a positive relationship with Minho River, and negative with temperature and Chl *a*. The second canonical axis showed a negative gradient with $NO_3$ and $Si(OH)_4$ and a negative relationship of these variables to *Chaetoceros* resting spores (ChaeRS). Conversely, *Leptocylindrus* resting spores (LepRS) were positively related with $NO_3$ and $Si(OH)_4$. The temporal distribution of the sediment trap samples showed that FW diatoms, benthic diatoms and Parsul occurred mainly during downwelling months while ChaeRS and LepRS were associated to upwelling periods (Fig. 8). Besides, this figure also discerns a location of LepRS towards the late summer and IPC periods.





## 5 Discussion

The siliceous microorganism fluxes, mostly represented by diatoms in both vegetative and resting spore stages, were strongly linked to biogenic silica fluxes and presented abrupt changes along the entire time series (Fig. 5). The observed temporal variability was determined by the regional hydrodynamic conditions at the NW Iberian inner continental margin, as

explained in detail by Zúñiga et al. (2016). These authors described how maximum particle fluxes occurring during downwelling periods were associated with resuspension processes mainly driven by wave action. This finding explains the a priori contradictory observation of maximum diatom fluxes during downwelling periods, when irradiance conditions were unfavourable and Chl *a* showed its minimum levels (Fig. 2, 3c and 5e). Conversely, during highly productive upwelling periods, even though biogenic silica fluxes were relatively low, diatoms contribution (including resting spores stage)

achieved maximum percentages (Fig. 5e and 5f). Therefore, our results confirm the major influence of the hydrodynamic conditions and seasonal productivity over the diatoms export to the surface sediment in this coastal upwelling system.

### 5.1. Sediment trap diatom assemblage as a tracer of allocthonous sources in sinking material

Maximum fluxes of benthic diatoms, whose natural habitat is the sediment interface, run parallel with higher wave heights during highly hydrodynamic downwelling periods (Fig. 2c and 6a), confirming that strong storms resuspended surface

sediments at the inner Iberian continental shelf (Dias et al., 2002; Vitorino et al., 2002. Jounneau et al., 2002, Oberle et al., 2014). Furthermore, stormy conditions were accompanied by intense Minho and Douro River discharges, resulting in lower salinity water lens at the sea surface, which had a significant effect over the water column thermohaline structure (Fig. 2d and 3a). The significant increase in freshwater diatoms associated to river runoff confirms how continental inputs constituted an additional source of terrestrial material to the inner continental shelf (Fig. 2c and 6c). Indeed, canonical analysis of

sediment trap samples revealed a high correlation between benthic and freshwater diatoms, corroborating the co-ocurrence of both resuspension processes and river discharges during downwelling periods (Fig. 8 and Table 2).
One additional evidence of resuspension resulted from the analysis of marine diatom assemblage collected in the sediment trap. *Paralia sulcata* was sporadically found in the water column diatom assemblage during the 2009-2012 studied years (Fig. 4c and Table 3). This meroplanktonic and shadow species, it is by contrast, common in the sediments and highly

contributed to the diatom fluxes during downwelling phases (Fig. 7c and Table 3). This observation can only be explained by the fact that *Paralia sulcata* is a robust and highly silicified species, relatively more resistant to dissolution processes, that gets enriched in the sediments. Consequently, and as pointed out by previously published sediment trap data from the adjacent Ría de Vigo (Bernárdez et al., 2010; Zúñiga et al., 2011), our data confirms it as a species that can be easily resuspended from the sediments under highly hydrodynamic conditions. This is, in fact, also exposed by the positive

relationship found between *Paralia sulcata* and benthic diatoms in the trap samples (Fig. 8 and Table 2). Also interesting is the positive correlation between freshwater and benthic diatoms to *Thalassiosira eccentrica* (Table 2), a species which is





known to occur in areas where nutrient input is continuous throughout the year, such as in areas influenced by river discharge (Moita, 1993, Abrantes and Moita, 1999).

## 5.2 Seasonal succession of diatom species during upwelling seasons: the imprint over the fossil diatom assemblage

During the studied period, the living diatom community was strongly linked to the seasonality revealed by environmental variables, with the highest abundances always recorded during upwelling favourable periods, when irradiance and water column characteristics promote favourable conditions for diatom growing (Fig. 2a and 3).

A detailed analysis of the marine diatom assemblage during upwelling productive seasons revealed that most living species linked to upwelling favourable conditions were not found (e.g. *Asterionellopsis glaciallis*, *Detonula pumila*, *Guinardia delicatula* and *Skeletonema costatum*) or appeared with a significant lower contribution (e.g. *Nitzschia* spp., *Pseudo-nitzschia* spp. and small centric) in the diatom assemblages on both the sediment trap and the surface sediment samples (Table 3). This observation points to selective dissolution processes acted over thin-walled, less silicified diatoms. As a result, the robust and heavily silicified frustules, not only have a ballast effect that promotes a faster arrival to the sediments, as have a higher preservation potential in the seafloor sediments. Indeed, diatoms assemblages in both sediment trap and surface sediment samples were mainly represented by the highly resistant *Chaetoceros* and *Leptocylindrus* spp. resting spores (Table 3). This fact confirms these diatom genera as a good sedimentary imprint of highly productive upwelling conditions. Moreover, the sink of *Chaetoceros* and *Leptocylindrus* spp. resting spores, occurring in close correlation with the dominance of both diatom groups in the water column assemblage during the upwelling periods (Table 3), brings new important information to previous works carried out along the Iberian margin which have only considered *Chaetoceros* spp. resting spores group as tracer of the coastal upwelling regime (Abrantes, 1988; Bao et al., 1997; Abrantes and Moita, 1999).

Certainly, canonical analysis performed for the sediment trap data, with *Chaetoceros* and *Leptocylindrus* spp. spores reflecting upwelling favourable conditions (positively positioned in CCA1), also showed the two genera differently placed with respect to significant environmental variables (Fig. 8). Overall, the sink of *Chaetoceros* spp. resting spores out from the photic zone was mostly associated to spring-summer periods (Fig. 8), revealing the onset of the upwelling seasons when irradiance conditions are favourable and persistent northerly winds lead to the upwelling of nutrient-rich subsurface ENACW waters on the shelf (Fig. 3, 4e and 4f) (Figueiras and Rios, 1993; Nogueira and Figueiras, 2005). Otherwise, *Leptocylindrus* spp. resting spores fluxes at the base of the photic layer were significantly associated to late-summer autumn (Fig. 8), marking highly productive upwelling periods (as shown by Chl *a* levels) when the relaxation of winds promotes water column stratification due to surface warming, and nutrients associated to upwelling (nitrates and phosphates) become depleted (Fig. 3, 4e and 4f) (Escaravage et al., 1999; Casas et al., 1999; Nogueira and Figueiras, 2005).

In summary, even though we do not dispose of a quantitative estimate of water column/sediment record preservation efficiencies due to strong resuspension processes remobilized bottom sediments at the inner NW Iberian continental shelf, our results expose an important role of *Chaetoceros* and *Leptocylindrus* spp. resting spores during the highly productive



upwelling intervals. Indeed, our data comprise important implications for paleoceanographic and paleoclimatic studies, since a careful evaluation of the *Chaetoceros* and *Leptocylindrus* spp. spores contribution to the total marine diatom assemblage present in the sediment records should allow the identification of persistent vs intermittent upwelling favourable winds, with *Leptocylindrus* spp. spores being more abundant in an intermittent upwelling mode.

## 5    Acknowledgements

The authors gratefully thank the crew of "R/V *Mytilus*" for their valuable help during the cruises. The authors also want to specially recognize specific lab work made from different members of both the Oceanography group from the Instituto de Investigaciones Marinas de Vigo (CSIC) and the Divisão de Geologia e Georecursos Marinhos from Instituto Português do Mar e da Atmosfera (IPMA). Special thanks to M. Zúñiga for his help in the writing process. This study was sponsored by

CAIBEX (CTM2007-66408-C02-01/MAR) and REIMAGE (CTM2011–30155–C03–03) projects funded by the Spanish Government, EXCAPA project (10MDS402013PR) supported by Xunta de Galicia, the EU FEDER funded projects RAIA (INTERREG 2009/2011-0313/RAIA/E) and RAIA.co (INTERREG 2011/2013-052/RAIA.co/1E) and the CALIBERIA project (PTDC/MAR/102045/2008) financed by Fundação para a Ciência e a Tecnologia (FCT-Portugal). D.Z. and E.S. were funded by a postdoctoral fellowship (Plan I2C) from Xunta de Galicia (Spain) and (SFRH/BPD/111433/2015) from FCT,

respectively. C.S. was funded by a doctoral grant from FCT (Portugal) (SFRH/BD/88439/2012).

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





Table 1. Environmental variables matrix Pearson correlations. Irrad: Irradiance; Temp: temperature; Sal: salinity; $N^2$: Brunt Väisälä frequency parameter; Chl $a$: Chlorophyll $a$; SPM: suspended particulate matter; POC: particulate organic carbon; UI: Upwelling index; Minho: Minho River discharge; Waves: Significant wave height in A Guarda (WANA$_G$) station. Main relationships are highlighted with grey background.

|  | Irrad | Temp | Sal | $N^2$ | Chl $a$ | SPM | POC | NO$_3$ | PO$_4$ | Si(OH)$_4$ | Oxygen | UI | Minho | Waves |
|---|---|---|---|---|---|---|---|---|---|---|---|---|---|---|
| **Irrad** | 1.00 | -0.394 | **0.487** | -0.0827 | **0.528** | -0.555 | 0.376 | 0.0167 | 0.0764 | -0.642 | 0.148 | **0.487** | -0.236 | -0.357 |
| **Temp** |  | 1.00 | -0.263 | **0.616** | -0.196 | 0.332 | 0.184 | -0.284 | -0.195 | 0.351 | -0.346 | -0.456 | 0.0961 | 0.0557 |
| **Sal** |  |  | 1.00 | -0.0509 | 0.228 | -0.645 | 0.348 | 0.0839 | 0.263 | -0.841 | -0.137 | **0.518** | -0.689 | -0.27 |
| **N$^2$** |  |  |  | 1.00 | -0.284 | 0.249 | 0.124 | -0.266 | 0.0559 | 0.0825 | -0.38 | -0.266 | -0.101 | -0.0122 |
| **Chl $a$** |  |  |  |  | 1.00 | -0.476 | **0.355** | 0.0608 | -0.0003 | -0.321 | 0.124 | 0.205 | 0.0433 | -0.272 |
| **SPM** |  |  |  |  |  | 1.00 | -0.104 | 0.153 | 0.0953 | **0.79** | -0.223 | -0.409 | **0.334** | **0.257** |
| **POC** |  |  |  |  |  |  | 1.00 | 0.24 | 0.341 | -0.201 | -0.39 | 0.0841 | -0.246 | -0.242 |
| **NO$_3$** |  |  |  |  |  |  |  | 1.00 | **0.871** | 0.287 | -0.677 | 0.182 | -0.106 | -0.252 |
| **PO$_4$** |  |  |  |  |  |  |  |  | 1.00 | 0.137 | -0.823 | 0.233 | -0.274 | -0.185 |
| **Si(OH)$_4$** |  |  |  |  |  |  |  |  |  | 1.00 | -0.297 | -0.466 | **0.523** | **0.240** |
| **Oxygen** |  |  |  |  |  |  |  |  |  |  | 1.00 | 0.0592 | 0.213 | 0.204 |
| **UI** |  |  |  |  |  |  |  |  |  |  |  | 1.00 | -.0394 | -0.116 |
| **Minho** |  |  |  |  |  |  |  |  |  |  |  |  | 1.00 | 0.139 |
| **Waves** |  |  |  |  |  |  |  |  |  |  |  |  |  | 1.00 |



Table 2 Sediment trap diatom species matrix Pearson correlations that appear in more than one sample with percentages higher than 2% of the total abundance. FW: freshwater diatoms; ChaetoRS: *Chaetoceros* spp. resting spores; Cos. mar: *Coscinodiscus marginatus*; Cos. rad: *Coscinodiscus radiatus*; LeptoRS: *Leptocylindrus* spp. resting spores; Nav: *Navicula* ; Nitzs.mar: *Nitzschi*a *marina*; Par.sulc: *Paralia sulcata*; Pse.Pun: *Pseudo-nitzschia pungens*; Thal.ecc: *Thalassiosira eccentrica*; Thal.nitzs: *Thalassionema nitzschioides*. Main relationships are highlighted with grey background.

| | FW | Benthic | ChaetoRS | Cos.mar | Cos.rad | LeptoRS | Nav.spp | Nitzs.mar | Par.sulc | Pse.pun | Thal.ecc | Thal.nitzs |
|---|---|---|---|---|---|---|---|---|---|---|---|---|
| **FW** | 1.00 | **0.482** | -0.0085 | -0.312 | 0.161 | 0.226 | -0.126 | 0.176 | 0.293 | 0.0318 | **0.382** | -0.0879 |
| **Benthic** | | 1.00 | -0.38 | 0.267 | 0.51 | -0.377 | 0.2 | 0.0623 | **0.671** | 0.159 | **0.428** | 0.0777 |
| **ChaetoRS** | | | 1.00 | -0.315 | -0.388 | -0.0163 | -0.147 | -0.389 | -0.493 | -0.318 | -0.337 | -0.214 |
| **Cos.mar** | | | | 1.00 | **0.416** | -0.137 | **0.517** | -0.138 | 0.201 | 0.15 | 0.182 | 0.253 |
| **Cos.rad** | | | | | 1.00 | 0.283 | 0.267 | 0.0427 | **0.487** | 0.0773 | 0.261 | 0.202 |
| **LeptoRS** | | | | | | 1.00 | -0.224 | -0.107 | -0.495 | -0.185 | -0.0972 | -0.173 |
| **Nav.spp** | | | | | | | 1.00 | -0.0424 | 0.196 | 0.18 | 0.074 | 0.174 |
| **Nitzs.mar** | | | | | | | | 1.00 | 0.173 | **0.547** | **0.338** | 0.079 |
| **Par.sulc** | | | | | | | | | 1.00 | 0.113 | 0.288 | 0.0748 |
| **Pse.pun** | | | | | | | | | | 1.00 | **0.412** | **0.561** |
| **Thal.ecc** | | | | | | | | | | | 1.00 | **0.451** |
| **Thal.nitzs.** | | | | | | | | | | | | 1.00 |





Table 3. Total abundance and relative contributions of the marine diatom assemblage preserved in the water column, sediment trap and surface sediments samples, respectively. SD: Standard deviation. Nitzs: *Nitzschia*; Pseudo-nitzs: *Pseudo-nitzschia*; Thal. nitzs: *Thalassionema nitzschioides*; Nav: *Navicula*. Small centric in both sediment trap and surface sediment samples includes: *Coscinodiscus marginatus, Coscinodiscus radiatus, Thalassiosira eccentrica*; Chaet. and Lepto. spp:

5  *Chaetoceros* and *Leptocylindrus* spp. cells and resting spores counted in the water column and sediment trap, respectively; Aster. glac: *Asterionellopsis glacialis*; Deton. pum: *Detonula pumila*, Guin. del: *Guinardia delicatula*, Skelet. cost: *Skeletonema costatum*. Most relevant species are highlighted in bold.

| | Total | Nitzs. spp. | Pseudo-nitzs. spp. | Thal. nitzs. | Small centric | Nav. spp. | *Paralia sulcata* | Chaeto. spp | Lept. spp | Aster. glac. | Deton. pum. | Guin. del. | Skelet. cost. |
|---|---|---|---|---|---|---|---|---|---|---|---|---|---|
| **WATER column** | | | | | | | | | | | | | |
| **Upwelling** | | | | | | | | | | | | | |
| cel mL$^{-1}$ (± SD) | 717 (1869) | 17 (50) | 25 (30) | 10 (17) | 486 (1766) | 1 (2) | 0 (0) | **72 (108)** | **126 (271)** | 3 (8) | 17 (15) | 11 (12) | 1 (2) |
| % (± SD) | | 3(4) | 16 (24) | 0.1 (0.4) | 20 (31) | 0.1 (0.4) | 0 (0) | **27 (28)** | **26 (33)** | 1 (1) | 3 (9) | 2 (4) | 2 (9) |
| **Downwelling** | | | | | | | | | | | | | |
| cel mL$^{-1}$ (± SD) | 34 (49) | 2 (2) | 12 (21) | 1 (1) | **10 (18)** | 0 (0) | 1 (1) | 1 (1) | 1 (1) | 1 (1) | 1 (0) | 0 (0) | 0 (0) |
| % (± SD) | | 12 (12) | 18 (28) | 2 (4) | **52 (25)** | 2 (4) | 2 (4) | 7 (10) | 1 (3) | 2 (3) | 2 (7) | 0 (0) | 0 (0) |
| **TRAP** | | | | | | | | | | | | | |
| **Upwelling** | | | | | | | | | | | | | |
| 10$^4$ # m$^{-2}$ d$^{-1}$ (± SD) | 591 (975) | 0.2 (0.1) | 1.1 (5.8) | 0.7 (1.6) | 0.5 (0.7) | 0.1 (0.3) | 5.3 (13) | **40.3 (87.2)** | **23.0 (49.9)** | | | | |
| % (± SD) | | 2 (1) | 6 (6) | 2 (4) | 2 (2) | 1 (1) | 14 (13) | **46 (25)** | **20 (22)** | | | | |
| **Downwelling** | | | | | | | | | | | | | |
| 10$^4$ # m$^{-2}$ d$^{-1}$ (± SD) | 2158 (4616) | 0.2 (0.5) | 0.9 (2.1) | 9.2 (2.1) | 1.3 (2.1) | 0.3 (0.5) | **31.2 (43.6)** | 25.6 (40.6) | 43.9 (111) | | | | |
| % (± SD) | | 1 (1) | 1 (1) | 2 (2) | 3 (2) | 1 (1) | **23 (12)** | 28 (19) | 24 (19) | | | | |
| **SEDIMENTS** | | | | | | | | | | | | | |
| **GeoB 11002-1** | | | | | | | | | | | | | |
| 10$^4$ # valves gr$^{-1}$ | 142 | 0 | 0 | 2 | 3 | 0 | **24** | **47** | **52** | | | | |
| % | | 0.2 | 0 | 2 | 2 | 0 | **17** | **33** | **37** | | | | |
| **GeoB 11003-2** | | | | | | | | | | | | | |
| 10$^4$ # valves gr$^{-1}$ | 65 | 0 | 0 | 2 | 1 | 0 | **7** | **26** | **26** | | | | |
| % | | 1 | 0 | 3 | 2 | 0 | **10** | **39** | **39** | | | | |



**Figure captions**

Fig. 1. Map of the NW Iberian Peninsula continental margin showing the position of the mooring line (RAIA) site. WANA hindcast reanalysis points 3027034 (WANA$_S$ off Cape Silleiro) and 1044067 (WANA$_G$ off A Guarda) from which wave data were obtained, location of the irradiance Cíes station (IR) and positions of the sediment cores GeoB11002-1 and GeoB11003-2 are also shown.

Fig. 2. Temporal series of (a) total irradiance at Cies station (IR), (b) upwelling index (-Q$_x$), (c) significant wave heights (Hs) obtained for propagation from the Silleiro and A Guarda WANA T points and, (d) Minho and Douro River discharges. Upwelling and downwelling periods are highlighted with white and grey bars, respectively, based on upwelling index and biogeochemical data presented in Zúñiga et al. (2016).

Fig. 3. Temporal series of (a) temperature (and water column integrated Brunt-Vaisälä frequency (N$^2$)), (b) nitrates (NO$_3$) and silicates (Si(OH)$_4$) content and, (c) Chl *a* concentration and diatoms abundance measured at 5 m water depth. Upwelling and downwelling periods are highlighted with white and grey bars, respectively, based on upwelling index and biogeochemical data presented in Zúñiga et al. (2016).

Fig. 4. Time series of diatom abundances (a, c, e and g) and assemblages (b, d, f, h) at 5 m water depth. Water column diatom species has been classified in order to easily compare them with fossil diatom assemblage from the sediment trap samples. Nitzschia spp: *Nitzschia longissima*; *Pseudo-nitzschia* spp: *Pseudo-nitzschia* cf. *delicatissima* and *Pseudo nitzschia* cf. *seriata*; Thal. nitzs: *Thalassionema nitzschioides*; small centric group: includes small centric diatoms and *Thalassiosira* spp. small; Navicula spp: *Navicula transitans* var. *derasa*; *Chaetoceros* (Ch.) spp: Ch. *curvisetus*, Ch. *socialis*, Ch. *didymus*, Ch. *laciniosus*, Ch. *decipiens* and small *Chaetoceros*; *Leptocylindrus* spp: *Leptocylindrus danicus*. Upwelling and downwelling periods are highlighted with white and grey bars, respectively, based on upwelling index and biogeochemical data presented in Zúñiga et al. (2016).

Fig. 5. Time series of (a) biogenic silica (BioSi), (c) total siliceous organisms and (e) diatoms (including diatom valves and spores) fluxes registered at RAIA station. Relative contribution of (b) biogenic silica to total mass flux, (d) silicoflagellates respect to total siliceous organisms and (f) resting spores to total diatoms flux are also presented. Upwelling and downwelling periods are highlighted with white and grey bars, respectively, based on upwelling index and biogeochemical data presented in Zúñiga et al. (2016).

Fig. 6. Time series of (a) benthic and (c) freshwater diatom fluxes (and relative contributions respect to total diatoms) (b, d) registered at RAIA station. Upwelling and downwelling periods are highlighted with white and grey bars, respectively, based on upwelling index and biogeochemical data presented in Zúñiga et al. (2016).

Fig. 7. Time series of marine diatoms fluxes (a, c, e) and assemblages (percentage respect to total marine diatoms) (b, d, f) registered at RAIA station. Fossil diatom species has been classified in three groups in order to compare them with water column diatom assemblage. *Nitzschia* spp: *Nitzschia marina*; *Pseudo-nitzschia* spp: *Pseudo-nitzschia pungens*; Thal. Nitzs: *Thalassionema nitzschioides*. Small centric group: includes *Coscinodiscus marginatus*, *Coscinodiscus radiatus* and

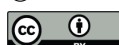

*Thalassiosira eccentrica*. Upwelling and downwelling periods are highlighted with white and grey bars, respectively, based on upwelling index and biogeochemical data presented in Zúñiga et al. (2016).

Fig. 8. RDA biplot results of the canonical ordination (only significant variables shown) for main fossil sediment trap diatom groups (freshwater (FW) diatoms, benthic diatoms, *Paralia sulcata* (Parsul), *Chaetoceros* spp. spores (ChaeRS) and

5  *Leptocylindrus* spp. spores (LepRS), and forward selected environmental variables (Chlorophyll *a* (Chl *a*), Temperature (Temp), nitrates ($NO_3$), silicates ($Si(OH)_4$) and Minho River flow). JFM: January-February-March, AMJ: April-May-June, JAS: July-August-September, OND: October-November-December.

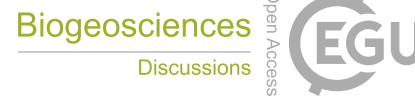



Figure 1

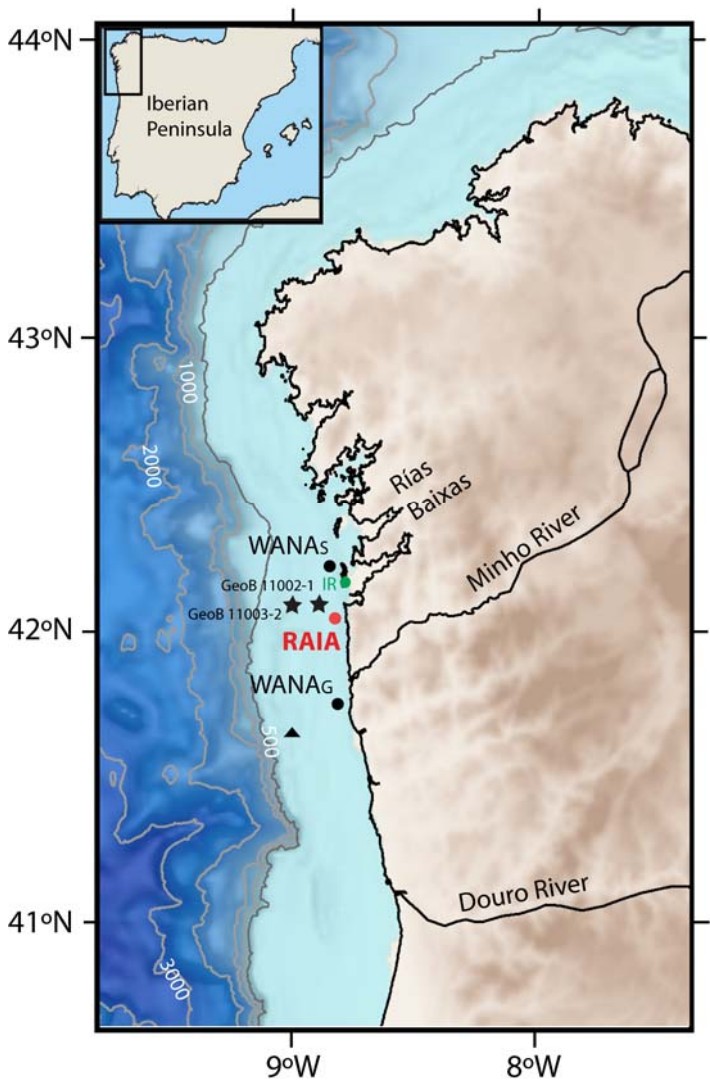





Figure 2



Figure 3

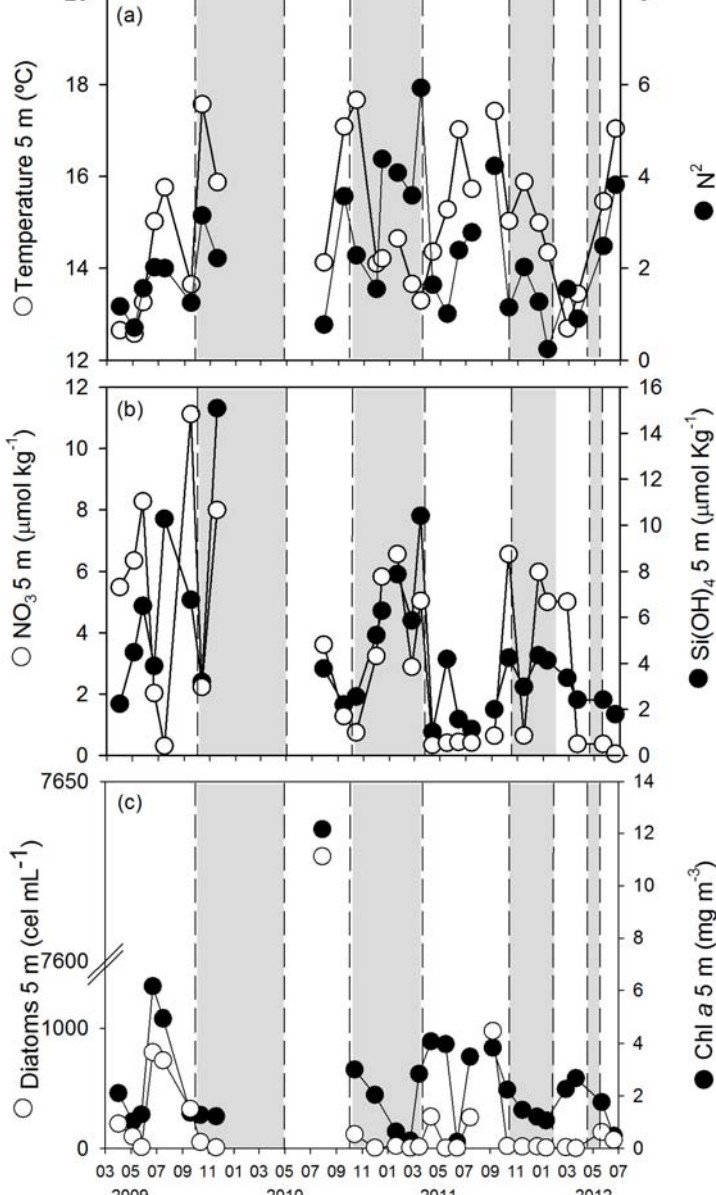



Figure 4





Figure 5



Figure 6



Figure 7







Figure 8

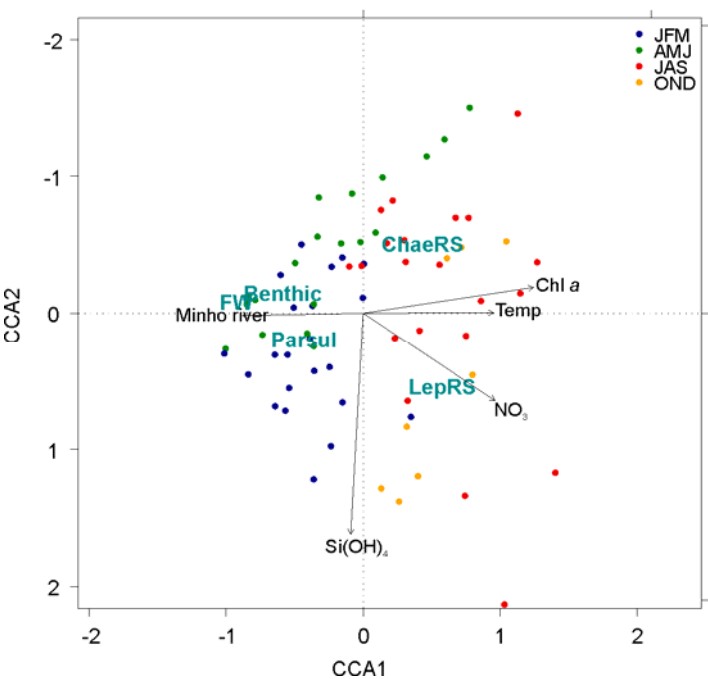



**Appendix A.** List of diatom species found in both the RAIA sediment trap and surficial sediment samples (Geo B 11002-1 and Geo 11003-2).

| Diatom species | Sediment trap | Sediments |
|---|---|---|
| 5 *Achnanthes brevipes* C. Agardh | X | X |
| *Achnanthes* sp. | X | |
| *Actinocyclus curvatulus* Janisch | X | X |
| *Actinocyclus octonarius* Ehrenberg | X | |
| *Actinocyclus* sp. | X | X |
| 10 *Actinoptychus senarius* (Ehrenberg) Ehrenberg | X | X |
| *Amphora gracilis* Ehrenberg | X | |
| *Amphora marina* T.V. Desikachary & P. Prema | X | |
| *Amphora* sp. | X | |
| *Anorthoneis excentrica* (Donkin) Grunow | X | |
| 15 *Asteromphalus flabellatus* (Brébisson) Greville | X | |
| *Asteromphalus* sp. | X | |
| *Aulacoseira* cf. *granulata* (Ehrenberg) Simonsen | X | X |
| *Aulacoseira* sp. | X | X |
| *Azpeitia neocrenulata* (S.L.VanLandingham) | X | |
| 20 *Azpeitia nodulifera* (A.Schmidt) G.A.Fryxell & P.A.Sims | X | |
| *Bacillaria paxillifera* (O.F. Müller) T.Marsson | X | |
| *Bacteriastrum hyalinum* Lauder | X | |
| *Caloneis* sp. | X | |
| *Campylodiscus incertus* A.W.F. Schmidt | X | |
| 25 *Campyloneis grevillei* (W.Smith) Grunow & Eulenstein | X | |
| *Campylosira cymbelliformis* Grunow ex Van Heurck | X | |
| *Cerataulus smithii* Ralfs ex Pritchard | X | |
| *Chaetoceros lorenzianus* Grunow | X | |
| *Chaetoceros* sp. (resting spores) | X | X |
| 30 *Chaetoceros* sp. | X | |
| *Cocconeis disculoides* (Hustedt) Stefano & Marino | X | |
| *Cocconeis guttata* Husted & Aleem | X | |
| *Cocconeis hoffmannii* Simonsen | X | |
| *Cocconeis neodiminuta* Krammer | X | |
| 35 *Cocconeis placentula* Ehrenberg | X | X |
| *Cocconeis pseudomarginata* Gregory | X | |
| *Cocconeis scutellum* Ehrenberg | X | X |
| *Cocconeis speciosa* Gregory | X | |
| *Cocconeis stauroneiformis* (W.Smith) H. Okuno | X | |
| 40 *Cocconeis* sp. | X | |
| *Coscinodiscus gigans* Ehrenberg | X | |
| *Coscinodiscus marginatus* Ehrenberg | X | X |
| *Coscinodiscus* cf. *oculus-iridis* (Ehrenberg) Ehrenberg | X | |
| *Coscinodiscus radiatus* Ehrenberg | X | X |
| 45 *Coscinodiscus* sp. | X | |
| *Ctenophora pulchell*a (Ralfs ex Kützing) | X | |
| *Cyclotella meneghiniana* Kützing | X | X |





|  | | X | |
|---|---|---|---|
|  | *Cyclotella plitvicensis* Husted | X | |
|  | *Cyclotella stelligera* Cleve & Grunow in Van Heurck | X | |
|  | *Cyclotella radiosa* (Grunow) Lemmermann | X | |
|  | *Cyclotella* sp. | X | |
| 5 | *Cyclostephanos* sp. | X | |
|  | *Cymbella affinis* Kützing | X | X |
|  | *Cymbela* sp. | X | |
|  | *Delphineis minutissima* (Husted) Simonsen | X | X |
|  | *Delphineis surirella* (Ehrenberg) G.W. Andrews | X | |
| 10 | *Detonula pumila* (Castracane) Gran | X | |
|  | *Dimeregramma minor* (Gregory) Ralfs ex Pritchard | X | |
|  | *Diploneis* cf. *bombus* (Ehrenberg) Ehrenberg | X | X |
|  | *Diploneis didymus* (Ehrenberg) Ehrenberg | X | |
|  | *Diploneis smithii* (Brébisson) Cleve | X | |
| 15 | *Diploneis* cf. *stroemii* Husted | X | |
|  | *Diploneis suborbicularis* (W.Gregory) Cleve | X | |
|  | *Diploneis weissflogii* (A.W.F.Schmidt) Cleve | X | |
|  | *Diploneis* sp. | X | |
|  | *Dytilum* sp | X | |
| 20 | *Encyonema* sp. | X | |
|  | *Epithemia* sp. | X | |
|  | *Eunotia* cf. *Pectinalis* (Kützing) Rabenhorst | X | |
|  | *Eunotia praerupta* (Grunow) | X | |
|  | *Eunotia* sp. | X | |
| 25 | *Fragilariforma constricta* (Ehrenberg) D.M.Williams & Round | X | |
|  | *Fragilaria crotonensis* (Kitton) Cleve & Möller | X | |
|  | *Fragilaria inflata* (Heiden) Hustedt | X | |
|  | *Fragilaria investiens* (W. Smith) Clever-Euler | X | |
|  | *Fragilaria schulzii* Brockmann | X | |
| 30 | *Fragilaria ulna* (Nitzsch) Lange-Bertalot) | X | X |
|  | *Fragilaria* sp. | X | |
|  | *Fragilariforma virescens* (Ralfs) D.M.Williams & Round | X | |
|  | *Gomphonema* sp. | X | X |
|  | *Gomphonema* cf. *acuminatum* (Ehrenberg) | X | |
| 35 | *Gomphonema* cf. *constrictum* (Ehrenberg) | X | |
|  | Gomphonema parvulum (Kützing) Kützing | X | |
|  | *Grammatophora angulosa* (Ehrenberg) | X | |
|  | *Grammatophora marina*    (Lyngbye) Kützing | X | X |
|  | *Grammatophora oceánica* (Ehrenberg) Cleve | X | |
| 40 | *Grammatophora* sp. | X | |
|  | *Grammatophora serpentina* (Ehrenberg) Hartley | X | |
|  | *Haslea* sp. | X | |
|  | *Hantzschia* sp. | X | |
|  | *Hemidiscus cuneiformis*    Wallich | X | |
| 45 | *Hemiaulus* sp. | X | |
|  | *Hyalodiscus scoticus* (Kützing) Grunow | X | |
|  | *Leptocylindrus* sp. (resting spores) | X | X |
|  | *Licmophora abbreviata* (C.Agardh) | X | |
|  | *Licmophora* sp. | X | |
| 50 | *Luticola mutica* (Kützing) D.G.Mann | X | |



|  |  | X |  |
|---|---|---|---|
|  | *Lyrella* sp. | X |  |
|  | *Melosira moniliformis* (O.F.Müller) C.Agardh | X |  |
|  | *Melosira varians* C.Agardh | X |  |
|  | *Melosira westii* W. Smith | X | X |
| 5 | *Melosira* sp. | X | X |
|  | *Navicula bacillum* Ehrenberg | X |  |
|  | *Navicula* cf. *cancellata* Donkin | X |  |
|  | *Navicula cincta* (Ehrenberg) Ralfs | X |  |
|  | *Navicula mutica* Kützing | X |  |
| 10 | *Navicula* cf. *pennata* A.Schmidt | X |  |
|  | *Navicula* sp | X |  |
|  | *Nitzschia angularis* W. Smith | X |  |
|  | *Nitzschia angustata* (W. Smith) Grunow | X |  |
|  | *Nitzschia longissima* (Brébisson) Ralfs | X |  |
| 15 | *Nitzschia macilenta* W. Gregory | X |  |
|  | *Nitzschia marina* Grunow | X | X |
|  | *Nitzschia panduriformis* W. Gregory | X |  |
|  | *Nitzschia umbonata* (Ehrenberg) H. Lange-Bertalot | X |  |
|  | *Nitzschia* sp. | X | X |
| 20 | *Odontella aurita* (Lyngbye) C.Agardh | X |  |
|  | *Odontella longicruris* (Greville) M.A.Hoban | X |  |
|  | *Odontella* sp. | X |  |
|  | *Opephora marina* (W. Gregory) Petit | X |  |
|  | *Opephora martyi* Héribaud | X | X |
| 25 | *Opephora* sp. | X |  |
|  | *Paralia sulcata* (Ehrenberg) Cleve | X | X |
|  | *Petroneis humerosa* (Brébisson ex W.Smith) Stickle & D.G.Mann | X |  |
|  | *Pinnularia borealis* Ehrenberg | X |  |
|  | *Pinnularia* sp. | X |  |
| 30 | *Pleurosigma elongatum* W. Smith | X |  |
|  | *Pleurosigma normanii* Ralfs in Pritchard | X |  |
|  | *Pleurosigma* sp. | X |  |
|  | *Pleurosira laevis* (Ehrenberg) Compère | X |  |
|  | *Porosira glacialis* (Grunow) Jørgensen | X |  |
| 35 | *Podosira stelliger* (Bailey) Mann | X | X |
|  | *Proboscia alata* (Brightwell) Sundström) | X |  |
|  | *Psammodiscus nitidus* (Gregory) Round & Mann | X |  |
|  | *Pseudo-nitzschia pungens* (Grunow ex Cleve) G.R.Hasle | X |  |
|  | *Pseudo-nitzschia seriata* (Cleve) H.Peragallo | X |  |
| 40 | *Raphoneis amphiceros* (Ehrenberg) Ehrenberg | X |  |
|  | *Rhabdonema arcuatum* (Lyngbye) Kützing | X |  |
|  | *Rhabdonema minutum* Kützing | X |  |
|  | *Rhabdonema* sp. | X |  |
|  | *Rhizosolenia bergonii* Peragallo | X |  |
| 45 | *Rhizosolenia hebetata* (Bailey) Gran | X |  |
|  | *Rhizosolenia* sp. | X |  |
|  | *Rhoicosphenia abbreviata* (C.Agardh) Lange-Bertalot | X |  |
|  | *Rhoicosphenia marina* (Kützing) M.Schmidt | X |  |
|  | *Roperia tesselata* (Roper) Grunow ex Pelletan | X | X |
| 50 | *Staurosirella pinnata* (Ehrenberg) D.M.Williams & Round | X |  |





|  |  |  |  |
|---|---|---|---|
| | *Staurosirella* sp. | X | |
| | *Stellarima stellaris* (Roper) G.R.Hasle & P.A.Sims | X | |
| | *Stephanodiscus astrea* (Ehrenberg) Grunow | X | |
| | *Stephanodiscus* sp. | X | X |
| 5 | *Stephanopyxis turris* (Greville) Ralfs in Pritchard | X | |
| | *Surirella* sp. | X | |
| | *Synedra gaillonii* (Bory) Ehrenberg | X | |
| | *Synedra* sp. | X | |
| | *Tabellaria fenestrata* (Lyngbye) Kützing | X | |
| 10 | *Tabellaria flocculos*a (Roth) Kützing | X | |
| | *Tabellaria* sp. | X | |
| | *Tetracyclus glan*s (Ehrenberg) F.W.Mills | X | |
| | *Thalassiosira eccentrica* (Ehrenberg) Cleve | X | X |
| | *Thalassiosira cf. leptopus* (Grunow) Hasle & G.Fryxell | X | X |
| 15 | *Thalassiosira lineata* Jousé | X | |
| | *Thalassiosir*a sp. | X | X |
| | *Thalassionema nitzschioides* (Grunow) Mereschkowsky | X | X |
| | *Toxarium undulatum* Bailey | X | |
| | *Trachyneis aspera* (Ehrenberg) Cleve 1894 | X | X |
| 20 | *Triceratium favus* Ehrenberg | X | |
| | *Tryblionella navicularis* (Brébisson) Ralfs | X | |