# Peer review of "Diatoms as a paleoproductivity proxy in the NW Iberian coastal upwelling system (NE Atlantic)"

_Biogeosciences, 2016_

## Referee Comment (RC1) · Anonymous Referee #1 · 21 Jun 2016

*The review comments on the manuscript entitled "Diatoms as paleoproductivity proxy in the NW Iberian coastal upwelling system (NE Atlantic)" by Zúñiga D, Santos C, Froján M, Salgueiro E, Rufino MM, De la Granda F, Figueiras FG, Castro CG, Abrantes F for possible publication in Biogeosciences.*

*Major comment*

The objectives of this paper is "How diatoms species determine primary production signal?" (quotation from Abstract), "to evaluate the use of marine diatoms as a paleoproductivity proxy for NW Iberian coastal upwelling system" (from Introduction). I understood the importance of resting spores and *Paralia sulcata* in diatom flora from sediment trap and sediment samples for paleoceanographic studies around the study area. It is meaningful to try to find the relationship among living diatom flora (biocoenosis), sinking flora obtained by sediment trap, and fossil assemblage in sediment (thanatocoenosis). The research results on diatom assemblage shows strong influence of lateral material advection and taphonomic bias due to seasonal downwelling event. As far as I saw the figures in this manuscript, it looks like that there is interannual variation in downwelling period length and intensity of rive discharge which probably reflects climate condition. The deciphering relationship between these events and diatom remains in sediment is another topic for further application of diatom fossils as a paleoenvironmental proxy. On the conclusion as the answer to objectives of this paper, it is a little bit uncertain for me to understand which of valve contents (valve number/dry g sediment) or % in total diatom valves is better to estimate paleo primary production signal. I feel that there are several points to be considered and/or corrected before the publication. My specific and editorial comments are as follows. Because I'm not a native English speaker, I did not care for the English style in this manuscript. I hope some of these comments are helpful to revise this manuscript.

*Specific and editorial comments*

2. Regional setting

p. 3, l. 11 "important source of terrestrial sediments to the inner shelf"

In addition to lithogenic material advection by river discharge, are there any significant influences of river discharge to nutrients for primary production and abundance of particulate organic matters in study area?

3.2. Water Column

p. 4, l. 9 "For diatoms counting … collected as 5m water depth"

Why water samples for diatom analysis were taken from 5 m depth? I'm not sure the water samples from 5m depth can be treated as a representative of diatom biocoenosis in water column. For example, would authors show vertical distribution of chlorophyll-a concentration during the monthly ship-board observation?

p. 4, l.12 "microorganisms were counted and identified"

      If resuspended dead specimens are included in the assemblage data from water samples for assemblage comparison with sediment trap and sediment samples, please mention it.

3.3. Sediment trap samples

p. 4, l.16 "from March 2009 to June 2012"

      As far as I see Figure 5, it looks like there were no deployment periods. If there is a blank of sampling period, please mention it in text or figure captions.

p. 4, l.16 "The trap was deployed at 35 m"

      I hope that the subsurface chlorophyll maximum was locating shallower than 35m throughout the sediment trap deployment period.

p. 4, l.19 "Only in exceptional cases… mooring tilts between 15-20°."

      When this event was observed?

p. 4, l. 32-p. 5, l.3 "Diatom flux was calculated as …where the flux F is expressed as number of valves $m^{-2}$ $d^{-1}$, …"

      How did authors count resting spores in sediment trap samples? The unit of resting spore flux in Fig. 7e is "number of spores $m^{-2}$ $d^{-1}$". In the result section (p.6, l. 27), Table 3, and Fig. 5e, total diatom flux including resting spores is expressed as "# $m^{-2}$ $d^{-1}$" which is different from the diatom flux unit explained here. The resting spore of *Chaetoceros* species listed in this study and *Leptocylindrus danicus* is composed of two valves. If total diatom flux including resting spore is expressed as the sum of vegetative cell "valve numbers" and "spore numbers" (not spore valve numbers), I think that the contribution of resting spore to total diatom flux will be underestimated and relative abundance of vegetative cells will be overestimated. If this is just a problem in expressive style in this manuscript, please revise the unit of spore abundance to avoid confusion. If resting spore contribution to total diatoms flux was actually underestimated in the dataset, re-analysis of CCA, replotting figures, data correction in tables 2 and 3 will be required.

p. 5, l.9 "a percentage higher than 2% of total abundance were considered for further analysis."

      Which species were applied to diatom analysis? please note it in appendix A table for encountered diatom taxa (for example, using bold font of "X").

p. 5, l.11 "42°10'N N,"

      Correct the duplication of "N"

p. 5, l.19 "main groups of diatoms (freshwater diatoms, benthic diatoms, …)"

      Is the category of benthic diatoms containing freshwater benthic and terrestrial species? If possible, would authors show which species are categorized into which group in Appendix A table, please?

4.1. Environmental conditions

p. 6, l.9-11 "Hydrographically, in a first phase … Later on, we differentiate the mixing period…"

Would authors show the first phase and later mixing period in the result figures?

p. 6, l. 14 "(~14 cel mL-1)"

cells?

p. 6, l. 14 "Small centric cells"

What is the definition of small cell size? The aim of this paper is to evaluate the marine diatom availability as a paleoproductivity while there is no discussion on the relationship between diatom abundance in cell volume (or carbon content) of each species and chlorophyll concentration (or primary production) throughout the manuscript. I see the high numerical dominance of *Chaetoceros* and *Leptocylindrus* in cell number in this study, and I agree their significant contribution among diatom species to primary production. However, some documentation on diatom contribution to primary production may be required in view point of cell volume or particulate carbon content for each species in studied samples. Because BioSi may not be primal component in siniking particles (Fig. 5b) and no information on other component in sinking particles, readers will not be sure that diatom is the most important contributor to primary production in the study area.

p. 7, l. 10 "Diatom abundance in GeoB 11002-1 … higher than in the offshore GeoB 11003-2"

What is main reason on the difference of diatom valve contents in surface sediment at two sites? On the cell contents and relative abundance of proxy species in sediment samples, which is better to use for paleoproduction proxy?

p. 7, l. 11 "# valves gr$^{-1}$"

The unit should be revised as # valves g$^{-1}$ based on International System of Units.

5. Discussion

p. 8, l. 9-10 "diatom contribution … achieved maximum percentage"

maximum percentage to what? To total biogenic opal flux or particulate organic carbon?

Comment: In general, coastal sediment sample except for varve sediment will be treated as an accumulation of settling particles for several-decadal years. In the case of studied sediments, diatom assemblage should mainly reflect the downwelling season because much of settling diatoms were supplied in downwelling season rather than upwelling season.

5.1. Sediment trap diatom assemblage as a tracer of allocthonous sources in sinking material

Comment: Are there any relationships between lithogenic particle fluxes and occurrence patterns of

freshwater and benthic diatom taxa if they are treated as a kind of proxies for sediment resuspension and lateral material advection? Can authors mention on time-series fluctuation of lithogenic particle flux in this paper?

5.2. Seasonal succession of diatom species during upwelling seasons: the imprint over the fossil diatom assemblage

Comment: As the answer to the objectives of this paper, it is a little bit uncertain for me which of valve contents and % of resting spores is better to use for the paleo productivity proxy? More additional and detail explanation may be required as my impression.

References

The style should be corrected to the format of Biogeosciences.

Table 1. "Main relationships"

How did authors define the main relationships? What is representing the data value in bold with darker gray background?

Table 2. "Pearson correlations"

It is unclear whether this Pearson correlation coefficient was calculated on diatom valve flux or relative abundance of each taxon in total diatom flux.

"Main relationships"

How did authors define the main relationships? $p < 0.05$?

Table 3.

What is the difference of 0 and blank in diatom flux data?

Fig. 4(a)

There is small arrow symbol located near the maximum flux of Small centric species. What does this arrow mean?

Fig. 4(e, f)

Are these plots including the data on resting spore of *Leptocylindrus* and *Chaetoceros*?

Fig. 5

The downwelling period in 2010-2011 is differ from Fig. 2. Please check and fix it.

Fig. 5( c)

The flux unit is a little bit uncertain. For example, "individuals $m^{-2}$ $d^{-1}$" may be better (in this case, diatom flux must be treated as cell number rather than valve number).

Fig. 6

The downwelling period in 2010-2011 is differ from Fig. 2. Please check and fix it.

Fig. 6(a) "Benthic"

Is this mean that benthic form of both marine-brackish water and freshwater species are included in this data set?

Fig. 7(a, b, e)

On the flux spike over the upper limit of vertical axis, I cannot find which species made the high flux.

---

## Referee Comment (RC2) · Anonymous Referee #2 · 19 Jul 2016

Zúniga and co-authors describe time series of diatoms fluxes and the species-specific composition of the assemblage in a high productive area off the northern NW Iberian coast and compare them to those preserved in nearby-by surface sediment samples. They studied the diatom community at 5 m (plankton samples), at 35 m (sediment trap) and at 75 m (slope bottom). Samples were intermittently collected between November 2008 and July 2012.

Based in previous observations, they identify upwelling and downwelling intervals between early 2009 and mid 2012. The length of these intervals was variable. As collected with sediment traps, highest biogenic silica (BSi) and diatom fluxes correspond to downwelling intervals. This was due to strong southwesterly storms and wave-driven

resuspension. The riverine input of terrestrial materials during fall and winter represents an additional source of particles and nutrients.

General comments:

- Several gaps, up to six months duration, interrupt the trap record. Neither further information nor detailed discussion on these issues are offered. Unfortunately, no continuous record for a full calendar year is presented. This makes the seasonal description and the seasonal comparison between years difficult.

- Results description is confusing and needs strong revision. I have listed below several sentences/statement, which need strong clarification.

- Discussion is driven by the idea of seasonal differences in the particle dynamics (upwelling / downwelling; see Figs 3-6), independent of the fact that for several of these intervals not enough data have been collected. Because of the lack of (continuous) observations, data tend to be over-interpreted.

- A native speaker should first revise the manuscript. Several language issues make the reading heavy. In particular, the English in the Discussion section is messy and tedious to read.

Specific comments:

Introduction: it is a smorgasbord of subjects and needs a more strong focus. We all know the critical role played by oceans in the global carbon cycle and the role of diatoms in fixation of CO2. However, how relevant is this for a manuscript dealing with a very regional signal under very particular oceanographic and atmospheric conditions?

P. 2, l. 10-12: stating that "Nevertheless, primary production reconstructions suffer from the uncertainty about how diatoms respond to particular environmental conditions, and how particular diatom species transfer primary production signal via exported and buried particles" is misleading. The authors ignores a huge body of published in the past 20 years addressing the issues of diatom signal in the uppermost water column

and is preservation in surface and downcore sediments. This needs revision.

P. 2, l. 31: is RAIA an acronym? In that case, what does it mean?

P. 3, 10: "During these hydrodynamic periods, this site ...", Which site? Revise. Same line: what is the yearly discharge of rivers Minho and Douro? Data are presented in Fig 2d, though there are hardly comments on this.

P. 4, l. 16: the RAIA was not "monitored", but waters overlying the station? Below in l. 17, was the trap deployed or within the photic zone and the mixed layer?

P. 5, l. 17: statistics were performed by using which diatom species data, relative or absolute abundance?

P. 5, l. 22-23: "resulted from water column data interpolation by considering sediment trap sample recovering intervals", difficult to understand, needs rephrsing.

P. 6, l. 6-11: this paragraph is copied from Zuniga et al., Continental Shelf Res. 123, p. 92.

P. 6, l. 11: I do not understand why "higher nutrients levels" occur when temperatures went down. Higher nutrient content might be due to stronger mixing or stronger eolian input/riverine discharge or a change in prevailing water masses, and might temporally match the occurrence of low(er) temperature. This needs revision.

P. 6, l. 14: What are these "small centric cells"? it is too vague and needs more accurate description.

P. 16, l. 15: "From May to October", this is true for 2010, though it seems to be March to Oct in 2009, while March to Sept in 2011 and Jan until July in 2012. Revise.

P. 6, l. 16: to me the use of "upwelling relaxation" and "promotion of upwelling of cold and nutrient rich ENACW" seems contradictory. Relaxation implies slackening of upwelling intensity (see e.g. Fraga et al., 1988, Continental Shelf Research 27, 349-361), hence if upwelling tends to weaken, why should influence the dynamics of

temperature?

P. 6, l. 18: a more through description of diatom values is needed here. Stating that "During the highly productive upwelling seasons, diatom abundances achieved maximum levels (up to 7629 cel mL-1)" only partially resemble results. The highest maximum is a unique event for the entire trap recorded period (most of values of total diatom flux are below 10*10(6) m-2 d-1).

P. 6, l. 24: there is own BSi value higher than 10% (Fig. 5b), revise.

P. 6, l. 25: I do not quite understand how the percentage of silicoflagellates was calculated.

P. 6, l. 27 and p. 7, l. 1: This statement "Maximum total diatom fluxes (up to up to 22.6 106 # m-2 d-1) were registered under downwelling conditions when benthic and freshwater diatoms fluxes became relevant" is wrong. Highest diatom fluxes (in the traps) do not temporally match highest relative contribution of benthic and freshwater diatoms. A gap of months is seen (compare Fig 5e and Fig 6b and d).

P. 7, l. 3 and 4: what do the authors intend to say here? This sentence is difficult to understand. Rephrase.

P. 7, l. 7: how these "mean seasonal values" were calculated? A table with the sampling intervals should be presented.

P. 8, l. 2: "The siliceous microorganism fluxes, mostly represented by diatoms", what about radiolarians? Low abundance? No present at all? Same line: it is not correct to state that "diatoms were strongly linked to biogenic silica fluxes and presented abrupt changes along the entire time series". In addition to the wrong grammar, it is usually interpreted the other way round: BSi fluxes are delivered by siliceous primary and secondary producers.

P. 8, l. l. 13: at which depth/s occur benthic diatoms?

P. 8, l. 17: where does the information on "lower salinity water lens at the sea surface" come from? Own observations? Fig 3a shows temperature data, no salinity measurements though.

P. 8, l. 24-30: this part of the Discussion section is convoluted and difficult to follow. It needs strong rephrasing. If, Paralia contributed to the 2010 downwelling interval and to the latest part of the 2011 downwelling intervals. Other than that, the lack of continuous records makes –at the least- this kind of generalizations. Although it is common in the two surface sediment studied, it relative contribution to the diatom preserved assemblage is always lower than that in the traps. Therefore, I do not understand why the authors state that "(Paralia sulcata) gets enriched in the sediments" (l. 27). Their interpretation is just erroneous.

P. 9, l. 5: which kind of highest abundances, relative or absolute?

P. 9, l. 7: writing "upwelling productive seasons" suggest that "upwelling non-productive season" occurred.

P. 9, l. 8-9: I do not understand where this statement comes from. . . it has been introduced in p. 6, l. 20, without referring to own observations or quoting previous work. Are the listed species typical of the coastal upwelling system along the Galician coast? Or are these species typical of any coastal upwelling system?

P. 9, l. 12: reference/s for ballast effect is needed here.

P. 9, l. 14: "Highly resistant" to what? I suppose to dissolution. . .

P. 9, l. 15-16: it is true that spores of Chaetoceros had high relative abundance during upwelling intervals. However, they do also contribute to the diatom assemblage during downwelling 2010. How can the authors explain this?

P. 9, l. 23: how deep is the photic zone at the trap site? See below, same page, l. 26.

P. 9, l. 23: "mostly associated to spring-summer periods (Fig. 8), revealing the onset of

the upwelling seasons", more accurateness is needed here. How can the authors associate spring-summer periods (six months?) with the onset of the upwelling season?

P. 9, l. 27-29: how can "highly productive upwelling periods (as shown by Chl a leves)" correspond to "relaxation of wind promoting water column stratification"? This is fully contradictory!

Comments on Figures

The dots representing data are too large. It is almost distinguishing sample resolution and actual values (e.g., Fig. 7b, 2012). Revise.

Figure 2b-d: these data were originally published in Zuniga et al., 2016, CSR.

Technical corrections

P. 6, l. 2, bracket missing.

―――――――――――――――――――――

---

## Author Comment (AC1) · 17 Sep 2016

**Responses to reviewer #1**

The review comments on the manuscript entitled "Diatoms as paleoproductivity proxy in the NW Iberian coastal upwelling system (NE Atlantic)" by Zúñiga D, Santos C, Froján M, Salgueiro E, Rufino, MM, De la Granda F, Figueiras FG, Castro CG, Abrantes F for possible publication in Biogeosciences.

Major comment
The objective of this paper is "How diatoms species determine primary production signal?" (quotation from Abstract), "to evaluate the use of marine diatoms as a paleoproductivity proxy for NW Iberian coastal upwelling system" (from Introduction). I understood the importance of resting spores and Paralia sulcata in diatom flora from sediment trap and sediment samples for paleoceanographic studies around the study area. It is meaningful to try to find the relationship among living diatom flora (biocoenosis), sinking flora obtained by sediment trap, and fossil assemblage in sediment (thanatocoenosis). The research results on diatom assemblage shows strong influence of lateral material advection and taphonomic bias due to seasonal downwelling event. As far as I saw the figures in this manuscript, it looks like that there is interannual variation in downwelling period length and intensity of rive discharge which probably reflects climate condition. The deciphering relationship between these events and diatom remains in sediment is another topic for further application of diatom fossils as a paleoenvironmental proxy. On the conclusion as the answer to objectives of this paper, it is a little bit uncertain for me to understand which of valve contents (valve number/dry g sediment) or % in total diatom valves is better to estimate paleo primary production signal. I feel that there are several points to be considered and/or corrected before the publication. My specific and editorial comments are as follows. Because I'm not a native English speaker, I did not care for the English style in this manuscript. I hope some of these comments are helpful to revise this manuscript.

**Response:** They agree with the idea that using diatoms fluxes as a paleoenvironmental proxy to infer downwelling conditions interannual variability is an interesting topic. However, they considered the need to focus the attention of the reader in the goal of the manuscript, the use of diatoms to infer the productivity in the past. On the other hand and in agreement with reviewer #1, they decided to better justify which is the best paleoproductivity production signal along the Iberian margin. To do that the authors modified some specific parts of the text. In this specific region, even the best diatom production signal is given by the total flux (valves/g), this regional calibration revealed that the export signal in terms of "absolute abundances" is distorted by strong resuspension processes during autumn-winter downwelling periods. This indicates that mean annual fluxes cannot be used, but only the upwelling period fluxes. In addition, this study states that the dominant species exported out from surface layer during upwelling productive seasons reflected the most prominent diatoms in the water column. All this means that although it is not possible to quantitatively determine the water column/sediment record preservation, the sediment diatom assemblages can be used to infer highly productive periods in this coastal

upwelling system. Otherwise, the identification of the water column environmental variables favourable for *Chaetoceros* and *Leptocylindrus* blooms (the two dominant species preserved in the sediment record through the formation of resting spores) provides relevant information for past environmental conditions.

Specific and editorial comments
2. Regional setting
p. 3, l. 11 "important source of terrestrial sediments to the inner shelf"
In addition to lithogenic material advection by river discharge, are there any significant influences of river discharge to nutrients for primary production and abundance of particulate organic matters in study area?

**Response:** In Zúñiga et al. (2016) the authors has exhaustively described water column and sediment trap biogeochemical data, showing among other things how intense river inputs during autumn-winter left their imprint as nitrate rich water lens at the sea surface. Even so, the authors consider that this additional input of nutrients during autumn-winter has no influence on primary production. Indeed, no relationship between nutrients associated to river inputs and Chl *a* concentration has been observed. On the contrary, Chl *a* contents were clearly related with the upwelling of cold and nutrients rich subsurface Eastern North Atlantic Central Water (ENACW) on the shelf during spring-summer periods.

The authors would also like to clarify that the term "terrestrial sediments" also include organic matter. This aspect has been also explained in detailed in Zúñiga et al. (2016) where the authors stated how "river discharges constitute an additional source of lithogenic and terrestrial organic matter to the sediment trap, resulting from the bulk of sediment washed out from the rivers onto the continental shelf". This information has not been included in the present manuscript to avoid duplication of already published information.

3.2. Water Column
p. 4, l. 9 "For diatoms counting … collected as 5m water depth" Why water samples for diatom analysis were taken from 5 m depth? I'm not sure the water samples from 5m depth can be treated as a representative of diatom biocoenosis in water column. For example, would authors show vertical distribution of chlorophyll-a concentration during the monthly ship-board observation?

**Response:** Vertical chloropyll *a* profiles were defined for the entire trap sampling period. This data has been previously published in Zúñiga et al., 2016 and for that reason the authors have decided not to present it here. Even so, a carefully data observation showed that seasonal Chl *a* variations were principally remarked at the topmost samples, with maximum values always registered at 5 m water depth (except July 2009). For this reason surface samples were selected

as the most representative of diatom biocoenosis in the water column. In any case, the authors have included additional information in the methods section to clarify this important aspect.

p. 4, l.12 "microorganisms were counted and identified"
If resuspended dead specimens are included in the assemblage data from water samples for assemblage comparison with sediment trap and sediment samples, please mention it.

**Response:** The samples recovered at 5 m water depth did not include "dead" specimens, but only "live cells" preserved with Lugol´s iodine solution.

3.3. Sediment trap samples
p. 4, l.16 "from March 2009 to June 2012" As far as I see Figure 5, it looks like there were no deployment periods. If there is a blank of sampling period, please mention it in text or figure captions.

**Response:** Following reviewer #1´s suggestion the authors have included additional information in methods section.

p. 4, l.16 "The trap was deployed at 35 m"
I hope that the subsurface chlorophyll maximum was locating shallower than 35m throughout the sediment trap deployment period.

**Response:** With the exception of June 2011 the subsurface chlorophyll maximum was always very shallow, well above the 35 m water depth. For further information please see Zúñiga et al. (2016).

p. 4, l.19 "Only in exceptional cases… mooring tilts between 15-20°."
When this event was observed?

**Response:** Mooring tilts of 15-20º were only observed when current velocities were higher than 25 cm s-1. Here, in this table (published in Zúñiga et al. (2016) it is possible to check how this situation only occurred sporadically.

| | Days | Average cm       s-1 | Maximum speed cm s-1 | Speed  <  20 (%) | Tilt < 5º (%) | Tilt < 10º (%) |
|---|---|---|---|---|---|---|
| Upwelling | 198 | 10.5 (6.8) | 41.0 | 94 | 76 | 92 |
| Poleward | 183 | 13.3 (10.6) | 62.7 | 85 | 57 | 82 |

| | | | | | | |
|---|---|---|---|---|---|---|
| Mixing | 63 | 15.4 (10.6) | 66.7 | 76 | 87 | 98 |

p. 4, l. 32-p. 5, l.3 "Diatom flux was calculated as …where the flux F is expressed as number of valves m-2 d-1, …" How did authors count resting spores in sediment trap samples? The unit of resting spore flux in Fig. 7e is "number of spores m-2 d-1". In the result section (p.6, l. 27), Table 3, and Fig. 5e, total diatom flux including resting spores is expressed as "# m-2 d-1" which is different from the diatom flux unit explained here. The resting spore of Chaetoceros species listed in this study and Leptocylindrus danicus is composed of two valves. If total diatom flux including resting spore is expressed as the sum of vegetative cell "valve numbers" and "spore numbers" (not spore valve numbers), I think that the contribution of resting spore to total diatom flux will be underestimated and relative abundance of vegetative cells will be overestimated. If this is just a problem in expressive style in this manuscript, please revise the unit of spore abundance to avoid confusion. If resting spore contribution to total diatoms flux was actually underestimated in the dataset, re-analysis of CCA, replotting figures, data correction in tables 2 and 3 will be required.

*Response:* *Chaetoceros* and *Leptocylindrus* spores are counted as valves (not spores). In order to clarify this aspect units expressed as # m$^{-2}$ d$^{-1}$ have been modified to valves m$^{-2}$ d$^{-1}$ in figures 5, 6 and 7. In addition, clarifications in both figure caption 5 and Table 3 have been made.

p. 5, l.9 "a percentage higher than 2% of total abundance were considered for further analysis." Which species were applied to diatom analysis? please note it in appendix A table for encountered diatom taxa (for example, using bold font of "X").

*Response:* Species used for sediment trap analysis were those species that appeared in more than one sample with a percentage higher than 2%. As suggested by reviewer #1 these species have been highlighted in bold in Appendix A.

p. 5, l.11 "42°10'N N,"
Correct the duplication of "N"
**Response:** Done

p. 5, l.19 "main groups of diatoms (freshwater diatoms, benthic diatoms, …)"
Is the category of benthic diatoms containing freshwater benthic and terrestrial species? If possible, would authors show which species are categorized into which group in Appendix A table, please?

**Response:** Main diatom ecological groups are shown in new version of Appendix A. The following ecological preferences were included: benthic, meroplanktonic, planktonic, coastal, open ocean, cosmopolitan, marine, marine to brackish, brackish to freshwater, brackish and freshwater.

3 4.1. Environmental conditions
p. 6, l.9-11 "Hydrographically, in a first phase … Later on, we differentiate the mixing period…"Would authors show the first phase and later mixing period in the result figures?

**Response:** The authors decided not to show hydrographic periods in the present manuscript since the objective of the paper is the comparison between upwelling and downwelling periods, as highlighted with the white and grey bars, respectively.

p. 6, l. 14 "(~14 cel mL-1)"cells?

**Response:** This is correct. It is referred to diatom cells. Modifications have been made in both figure 3 and 4.

p. 6, l. 14 "Small centric cells" What is the definition of small cell size? The aim of this paper is to evaluate the marine diatom availability as a paleoproductivity while there is no discussion on the relationship between diatom abundance in cell volume (or carbon content) of each species and chlorophyll concentration (or primary production) throughout the manuscript. I see the high numerical dominance of Chaetoceros and Leptocylindrus in cell number in this study, and I agree their significant contribution among diatom species to primary production. However, some documentation on diatom contribution to primary production may be required in view point of cell volume or particulate carbon content for each species in studied samples. Because BioSi may not be primal component in sinking particles (Fig. 5b) and no information on other component in sinking particles, readers will not be sure that diatom is the most important contributor to primary production in the study area.

**Response:** Small centric cells are referred to centric diatoms that for their size were not possible to be identified to the species level. In order to clarify this important aspect, additional information has been included in the methods section, figure caption 4 and Table 3.
and Table 3

The authors really agree with reviewer #1 in the sense that most important aspects to be resolved when analysing diatoms´ export and accumulation in sediment records is the fact to estimate how much of the particulate organic carbon attributed to primary production is susceptible to be transported to deep waters through these tiny organisms. In this frame, even

long-term sediment traps are extremely important to analyze seasonal diatom-derived organic carbon export rates to the deep, they do not provide us such a quantitative analysis since diatoms preserved in sediment trap samples are only represented by their valves, not by their cells. In this context and considering that water column diatoms are counted as "living cells" but sediment trap diatoms are numbered as "dead valves" the authors considered that evaluation of diatoms as a paleoproductivity proxy by analysing long-term sediment trap samples may only be carried out in terms of abundance.

The authors would also like to remark that they do not consider diatoms as the most important contributor to primary production in this margin. Indeed, there are some studies in the region that show how other phytoplankton groups are also important contributors to primary production. One good example is the manuscript from Espinoza-Gonzalez et al. (2012) entitled "Autotrophic and heterotrophic microbial plankton biomass in the NW Iberian upwelling: seasonal assessment of metabolic balance". Even so, the authors decide to evaluate the use of diatoms as a paleoproductivity proxy due to the capacity of their silica frustules to be preserved in the sediment record. They are frequently used in this margin to estimate productivity in the past but no attempt had yet been made to evaluate their abundance and assemblage composition in the photic zone and its transfer to the sediments up to this study.

The authors are aware that they have not estimated primary production and organic carbon export via diatoms, but comply with the objective by providing relevant information on how environmental conditions promote different diatom abundances and determine the blooming species as well as their preservation potential in the surface sediments.

p. 7, l. 10 "Diatom abundance in GeoB 11002-1 … higher than in the offshore GeoB 11003-2" What is main reason on the difference of diatom valve contents in surface sediment at two sites? On the cell contents and relative abundance of proxy species in sediment samples, which is better to use for paleoproduction proxy?

**Response:** Higher diatom abundances in the surface sediment at GeoB 11002-1 onshore station responded to primary production signal in the photic layer. Even not presented here the authors have evaluated Chl *a* contents through a transect perpendicular to the coast during almost the entire sampling period. They observed how seasonal Chl *a* at the surface productive layer is intensified close to the coast, in agreement with diatom´s abundance in surface sediment samples. This confirms the use of valves/g as a good indicator of diatom´s production in the photic zone. In that sense, since onshore surface sediment sample is closest to the RAIA station the authors consider it better to reflect the conditions at this site and will be the only one used in the new version of the manuscript.

p. 7, l. 11 "# valves gr-1" The unit should be revised as # valves g-1 based on International System of Units.

**Response:** Corrections have been made in the new version of the manuscript.

5. Discussion
p. 8, l. 9-10 "diatom contribution … achieved maximum percentage"
maximum percentage to what? To total biogenic opal flux or particulate organic carbon? Comment: In general, coastal sediment sample except for varve sediment will be treated as an accumulation of settling particles for several-decadal years. In the case of studied sediments, diatom assemblage should mainly reflect the downwelling season because much of settling diatoms were supplied in downwelling season rather than upwelling season.

**Response:** Following previous reviewer #1´s comments the whole paragraph has been modified.

5.1. Sediment trap diatom assemblage as a tracer of allocthonous sources in sinking material
Comment: Are there any relationships between lithogenic particle fluxes and occurrence patterns of freshwater and benthic diatom taxa if they are treated as a kind of proxies for sediment resuspension and lateral material advection? Can authors mention on time-series fluctuation of lithogenic particle flux in this paper?

**Response:** The authors agree that a comparison of the diatom assemblage and the lithogenic particle flux would help to justify it as a tracer of allocthonous sources. Thus, part of the text has been modified in the new version of the manuscript.

5.2. Seasonal succession of diatom species during upwelling seasons: the imprint over the fossil diatom assemblage Comment: As the answer to the objectives of this paper, it is a little bit uncertain for me which of valve contents and % of resting spores is better to use for the paleo productivity proxy? More additional and detail explanation may be required as my impression.

**Response:** After reading with attention the whole manuscript the authors agree with reviewer #1 that more explanation is required in relation with which diatom proxy needs to be used in order to infer the imprint of the water column diatom assemblage over the fossil diatom record. As explained in detail above, modifications in both abstract and discussion section have been made in the new version of the manuscript in order to clarify this fundamental aspect.

References

The style should be corrected to the format of Biogeosciences.

**Response:** Done

Table 1. "Main relationships" How did authors define the main relationships? What is representing the data value in bold with darker gray background?

**Response:** After carefully reading the paper the authors have decided to eliminate both the grey background and data values in bold. They consider the use of this format does not provide of relevant information for data discussion.

Table 2. "Pearson correlations" It is unclear whether this Pearson correlation coefficient was calculated on diatom valve flux or relative abundance of each taxon in total diatom flux. "Main relationships" How did authors define the main relationships? $p < 0.05$?

**Response:** New information related with Pearson correlation coefficients has been included in the new version of the manuscript.

With regards main relationships and p-values the authors would like to state that the main suspicious relationships on a preliminary analysis correspond to variables with absolute correlation coefficient higher than 50%. Still, the selection procedure on this type of analysis is not mathematical, but the result of an interactive procedure to reach a set of variables with highest correlations among them and most representative of all sampled variables. The ultimate instance to test if this set is proper for further analysis is the variance inflation factor, as it is explained in the text.

Table 3.
What is the difference of 0 and blank in diatom flux data?

**Response:** Corrected.

Fig. 4(a)
There is small arrow symbol located near the maximum flux of Small centric species. What does this arrow mean?

**Response:** The arrow with the corresponding number was used to show small cells abundance in July 2010 because it was out of scale. Additional information was added to the figure caption.

Fig. 4(e, f)
Are these plots including the data on resting spore of Leptocylindrus and Chaetoceros?

**Response:** For the water column samples, *Leptocylindrus* and *Chaetoceros* spp. data do not include resting spores since they are difficult to observe. *Leptocylindrus* and *Chaetoceros* spp. are only referred to live cells as shown in figure caption 4.

Fig. 5. The downwelling period in 2010-2011 is differ from Fig. 2. Please check and fix it.

**Response:** All figures have been exhaustively checked.

Fig. 5( c) The flux unit is a little bit uncertain. For example, "individuals m-2 d-1" may be better (in this case, diatom flux must be treated as cell number rather than valve number).

**Response:** Modified. Check comment immediately above.

Fig. 6
The downwelling period in 2010-2011 is differ from Fig. 2. Please check and fix it.

**Response:** Done

Fig. 6(a) "Benthic" Is this mean that benthic form of both marine-brackish water and freshwater species are included in this data set?

**Response:** No. In this particular case, fresh water benthic species were included in the fresh water group and the marine benthic were considered as benthic. As explained above, additional ecological information was included in new Appendix A.

Fig. 7(a, b, e)
On the flux spike over the upper limit of vertical axis, I cannot find which species made the high flux.

**Response:** The figure has been modified in the new version of the manuscript by adding coloured circles to the numbers.

---

## Author Comment (AC2) · 17 Sep 2016

**Responses to reviewer #2**

Zúniga and co-authors describe time series of diatoms fluxes and the species-specific composition of the assemblage in a high productive area off the northern NW Iberian coast and compare them to those preserved in nearby-by surface sediment samples. They studied the diatom community at 5 m (plankton samples), at 35 m (sediment trap) and at 75 m (slope bottom). Samples were intermittently collected between November 2008 and July 2012. Based in previous observations, they identify upwelling and downwelling intervals between early 2009 and mid 2012. The length of these intervals was variable. As collected with sediment traps, highest biogenic silica (BSi) and diatom fluxes correspond to downwelling intervals. This was due to strong southwesterly storms and wave-driven resuspension. The riverine input of terrestrial materials during fall and winter represents an additional source of particles and nutrients.

General comments:
- Several gaps, up to six months duration, interrupt the trap record. Neither further information nor detailed discussion on these issues are offered. Unfortunately, no continuous record for a full calendar year is presented. This makes the seasonal description and the seasonal comparison between years difficult.

- Results description is confusing and needs strong revision. I have listed below several sentences/statement, which need strong clarification.

- Discussion is driven by the idea of seasonal differences in the particle dynamics (upwelling / downwelling; see Figs 3-6), independent of the fact that for several of these intervals not enough data have been collected. Because of the lack of (continuous) observations, data tend to be over-interpreted.

- A native speaker should first revise the manuscript. Several language issues make the reading heavy. In particular, the English in the Discussion section is messy and tedious to read.

**Response:** The authors are really aware that the existence of gaps in the time series may mask some particular oceanographic features. However, the authors do not consider that the interruption of the trap record could lead to a misunderstanding of the seasonal patterns, since those repeatedly appear along the time series. They would like to emphasize that when they statistically analysed the time series in terms of seasonality, one of the main issues taken into account was to be confident of the existence of samples collected during each oceanographic season for every year. With the exception of year 2010, the authors found some general patterns in diatoms fluxes as being characteristics of each of the oceanographic seasons described by Zúñiga et al., (2016). Statistical analyses also corroborated our finding.

Results description has been revised and the authors hope it to be tighter now.

Following reviewer #2 suggestion, English has been revised. The authors hope that reviewer #2 will appreciate the improvements of the manuscript.

Specific comments:
Introduction: it is a smorgasbord of subjects and needs a more strong focus. We all know the critical role played by oceans in the global carbon cycle and the role of diatoms in fixation of CO2. However, how relevant is this for a manuscript dealing with a very regional signal under very particular oceanographic and atmospheric conditions?

**Response:** The authors agree with reviewer #2 that most people know the critical role played by the ocean in the global ocean carbon cycle. Though, in this first paragraph the authors not only asserted this idea but also highlight the importance of diatoms in a context of ocean carbon cycle as a key issue to introduce the manuscript. Furthermore, the authors find it important to justify the importance of these studies in regional coastal upwelling systems. This study(jointly with previous ones about diatom fluxes in coastal regions) is fundamental not only because these regions are the most productive regions in the global ocean but also because they are characterized by a strong seasonality both in terms of hydrodynamic and biological processes. Therefore, these areas are fundamental for paleoceanographic studies since they are likely to preserve relevant information on past productivity conditions and its causes.

In any case, first and second paragraphs of the introduction have been modified to clearly state the aim of the manuscript.

P. 2, l. 10-12: stating that "Nevertheless, primary production reconstructions suffer from the uncertainty about how diatoms respond to particular environmental conditions, and how particular diatom species transfer primary production signal via exported and buried particles" is misleading. The authors ignores a huge body of published in the past 20 years addressing the issues of diatom signal in the uppermost water column and is preservation in surface and downcore sediments. This needs revision.

**Response:** Paragraph has been rewritten. Please see comment immediately above.

P. 2, l. 31: is RAIA an acronym? In that case, what does it mean?

**Response:** RAIA is not an acronym; it is the local name for the northern border between Spain and Portugal. The term 'RAIA' is also the name of a EU FEDER funded project (INTERREG 2009/2011; 0313/RAIA/E). Our mooring line deployments were partially funded by this project as mentioned in the acknowledgments.

P. 3, 10: "During these hydrodynamic periods, this site ...", Which site? Revise. Same line: what is the yearly discharge of rivers Minho and Douro? Data are presented in Fig 2d, though there are hardly comments on this.

**Response:** The sentence has been modified in the new version of the manuscript.

P. 4, l. 16: the RAIA was not "monitored", but waters overlying the station?

**Response:** The sentence has been modified.

Below in l. 17, was the trap deployed or within the photic zone and the mixed layer?

**Response:** The sediment trap was deployed at the base of the photic zone, located on average at 35 m water depth. This information was obtained making use of a Secchi disk.

P. 5, l. 17: statistics were performed by using which diatom species data, relative or absolute abundance?

**Response:** The authors agree with reviewer #2 that additional information is required. The sentence has been modified in the new version of the manuscript.

P. 5, l. 22-23: "resulted from water column data interpolation by considering sediment trap sample recovering intervals", difficult to understand, needs rephasing.

**Response:** The sentence has been modified.

P. 6, l. 6-11: this paragraph is copied from Zuniga et al., Continental Shelf Res. 123, p. 92.

**Response:** The authors do not agree with reviewer #2. Hydrographic features are the same than those described by Zúñiga et al. (2016) because both studies were carried out over the same samples but the paragraph was not copied.

In Zúñiga et al. (2016) the paragraph is as follow:

*Prevailing south-westerly winds (negative –Qx values) were registered from October to April-May causing strong downwelling conditions. (Fig. 3(a)). Such conditions were accompanied by strong SW storms responsible for significantly high wave heights (HS) (up to 9.1 m)(Fig. 3(c)). Wave action lead to resuspension of bottom sediments as reflected by the significant increase in deep water column turbidity under stormy conditions(Fig. 4(e)). At the same time, intense Minho and Douro river discharges (4400 m3 s_1), left their imprint as low salinity (35.6) and relatively high nitrate content water lens at sea surface (Fig. 4b, e).*

In the present manuscript the paragraph is:

*From October to April-May, the NW Iberian margin was generally characterized by the prevalence of low irradiance levels and south-westerly winds as shown by the negative -Qx values (Fig. 2a and 2b). This downwelling season was also accompanied of strong SW storms promoting wave heights frequently higher than 4 m and intense Minho and Douro River discharges (Fig. 2c and 2d).*

P. 6, l. 11: I do not understand why "higher nutrients levels" occur when temperatures went down. Higher nutrient content might be due to stronger mixing or stronger eolian input/riverine discharge or a change in prevailing water masses, and might temporally match the occurrence of low(er) temperature. This needs revision.

**Response:** The sentence has been modified.

P. 6, l. 14: What are these "small centric cells"? it is too vague and needs more accurate description.

**Response:** Small centric cells are referred to centric diatoms that for their size were not possible to be identified to the species level. In order to clarify this important aspect, additional information has been included in the methods section, figure caption 4 and table 3.

P. 6, l. 15: "From May to October", this is true for 2010, though it seems to be March to Oct in 2009, while March to Sept in 2011 and Jan until July in 2012. Revise.

**Response:** The sentence has been modified in order to be coherent with first paragraph of section 4.1.

P. 6, l. 16: to me the use of "upwelling relaxation" and "promotion of upwelling of cold and nutrient rich ENACW" seems contradictory. Relaxation implies slackening of upwelling

intensity (see e.g. Fraga et al., 1988, Continental Shelf Research 27, 349-361), hence if upwelling tends to weaken, why should influence the dynamics of temperature?

**Response:** From March-April to October, northerly winds are responsible for the upwelling of cold and nutrient rich waters on the NW Iberian continental shelf. In this oceanographic context, it is frequent that upwelling favourable winds persist during spring months. However, during summer months, a series of upwelling-relaxation cycles are more characteristic of this study area. This means that during summer, we observe periods of intense ENACW upwelling and periods of winds relaxation that promote water column stratification. In any case this sentence has been modified to avoid confusion.

P. 6, l. 18: a more through description of diatom values is needed here. Stating that "During the highly productive upwelling seasons, diatom abundances achieved maximum levels (up to 7629 cel mL-1)" only partially resemble results. The highest maximum is a unique event for the entire trap recorded period (most of values of total diatom flux are below 10*10(6) m-2 d-1).

**Response:** The text on line 18 only refers to water column diatom abundances, not to the total diatom flux record.

P. 6, l. 24: there is own BSi value higher than 10% (Fig. 5b), revise.

**Response:** BioSi % is indeed higher than 10% by some decimal points, but values were all rounded to the unit.

P. 6, l. 25: I do not quite understand how the percentage of silicoflagellates was calculated.

**Response:** Percentage of silicoflagellates was calculated in relation to the total of siliceous microorganisms. This information is included in figure caption 5.

P. 6, l. 27 and p. 7, l. 1: This statement "Maximum total diatom fluxes (up to up to 22.6 106 # m-2 d-1) were registered under downwelling conditions when benthic and freshwater diatoms fluxes became relevant" is wrong. Highest diatom fluxes (in the traps) do not temporally match highest relative contribution of benthic and freshwater diatoms. A gap of months is seen (compare Fig 5e and Fig 6b and d).

**Response:** The authors are comparing fluxes therefore you should compare Fig 5e to Figs 6a and c. Total diatoms and spores showed a major flux peak at about one month delay from the beginning of the period and a second smaller peak flux towards the end of the downwelling

period. Both freshwater and benthic diatoms show the same two peaks with similar fluxes. Anyway, what the sentence labelled as wrong by the reviewer want to highlight is that "all three groups show maximum fluxes during downwelling periods". Furthermore, even in terms of relative percent contribution to the diatom assemblage, the higher contribution of both the freshwater and the benthic groups occurs during downwelling periods.

P. 7, l. 3 and 4: what do the authors intend to say here? This sentence is difficult to understand. Rephrase.

**Response:** Sentence has been modified in the new version of the manuscript.

P. 7, l. 7: how these "mean seasonal values" were calculated? A table with the sampling intervals should be presented.

**Response:** Sampling intervals are those from the figures. The authors considered not necessary to show sampling intervals in this manuscript since they were presented in detail in Zúñiga et al. (2016) and they do not provide additional information to the manuscript. Mean seasonal values have been calculated considering all samples recovered during upwelling periods and all samples recovered during downwelling periods.

*P. 8, l. 2: "The siliceous microorganism* fluxes, mostly represented by diatoms", what about radiolarians? Low abundance? No present at all? Same line: it is not correct to state that "diatoms were strongly linked to biogenic silica fluxes and presented abrupt changes along the entire time series". In addition to the wrong grammar, it is usually interpreted the other way round: BSi fluxes are delivered by siliceous primary and secondary producers.

**Response:** Radiolarians are not abundant in shallow coastal waters and they were not observed in the trap samples nor in the surface sediment samples.

P. 8, l. l. 13: at which depth/s occur benthic diatoms?

**Response:** Benthic diatoms as photosynthetic organism can occur in the sediments down to the depth of the photic layer, as such it varies from region to region.

P. 8, l. 17: where does the information on "lower salinity water lens at the sea surface" come from? Own observations? Fig 3a shows temperature data, no salinity measurements though.

**Response:** Salinity data is not presented in this figure and referred to Zúñiga et al. (2016). The sentence has been modified in the new version of the manuscript.

P. 8, l. 24-30: this part of the Discussion section is convoluted and difficult to follow. It needs strong rephrasing. If, Paralia contributed to the 2010 downwelling interval and to the latest part of the 2011 downwelling intervals. Other than that, the lack of continuous records makes –at the least- this kind of generalizations. Although it is common in the two surface sediment studied, it relative contribution to the diatom preserved assemblage is always lower than that in the traps. Therefore, I do not understand why the authors state that "(Paralia sulcata) gets enriched in the sediments" (l. 27). Their interpretation is just erroneous.

**Response:** Enriched in the sediments relatively to their water column abundance. Their presence in the traps only occurs when resuspension of bottom sediments occurred. *Paralia* is well known as a resistant planktonic species which abundance has been found to increase in the sediments comparatively to their occurrence in the water column in trap studies off NW Africa, California, Central Chile and others. Anyway, the sentence has been rewritten in order to clarify.

P. 9, l. 5: which kind of highest abundances, relative or absolute?

**Response:** Absolute

P. 9, l. 7: writing "upwelling productive seasons" suggest that "upwelling non-productive season" occurred.

**Response:** The authors do not agree with reviewer #2. In this context, the term "upwelling productive season" is only used to provide the reader with regards upwelling periods, specifying that upwelling seasons are typically highly productive. Indeed, this term is frequently used in works published in the study area.

P. 9, l. 8-9: I do not understand where this statement comes from: : : it has been introduced in p. 6, l. 20, without referring to own observations or quoting previous work. Are the listed species typical of the coastal upwelling system along the Galician coast? Or are these species typical of any coastal upwelling system?

**Response:** This statement refers to our own observations. Indeed, the authors make reference to Table 3 where the data is presented, for easier comparison. The authors would also like to

underline that even most species are common to coastal upwelling systems, the listed species are known to be typical of the NW Iberian peninsula coastal upwelling system.

P. 9, l. 12: reference/s for ballast effect is needed here.

**Response:** Done

P. 9, l. 14: "Highly resistant" to what? I suppose to dissolution:

**Response:** Sentence has been modified in order to be more precise.

P. 9, l. 15-16: it is true that spores of Chaetoceros had high relative abundance during upwelling intervals. However, they do also contribute to the diatom assemblage during downwelling 2010. How can the authors explain this?

**Response:** By winter resuspension of bottom sediments containing what is delivered to them during the summer

P. 9, l. 23: how deep is the photic zone at the trap site? See below, same page, l. 26.

**Response:** The sediment trap was deployed at the base of the photic zone, located on average at 35 m water depth. This information was corroborated making use of a Secchi disk during monthly cruises.

P. 9, l. 23: "mostly associated to spring-summer periods (Fig. 8), revealing the onset of the upwelling seasons", more accurateness is needed here. How can the authors associate spring-summer periods (six months?) with the onset of the upwelling season?

**Response:** From an oceanographic point of view, different phases during the upwelling season may be observed in the NW Iberian margin. There is a first phase that mainly occurred between late spring and early summer, characterized by the persistence of upwelling favourable winds, and a second phase, associated to late summer-early autumn when a series of upwelling relaxation cycles occurred. During this second phase, when winds relaxation are sustained in time favour water column stratification. Sentence has been modified in the new version of the manuscript to avoid confusion.

P. 9, l. 27-29: how can "highly productive upwelling periods (as shown by Chl a levels)" correspond to "relaxation of wind promoting water column stratification"? This is fully contradictory!

**Response:** Please see response above.

Comments on Figures
The dots representing data are too large. It is almost distinguishing sample resolution and actual values (e.g., Fig. 7b, 2012). Revise.

**Response:** The authors agree with reviewer 2. However, even they agree with the fact that some of the data remain hidden due to values for the different species are close to zero, after several test they decided not to modify the figure. They consider reduction of data size difficult graphics reading. Indeed, even they are aware of this detail in Figure 7b they consider data used for the discussion section are clearly shown.

Figure 2b-d: these data were originally published in Zuniga et al., 2016, CSR.

**Response:** Even these data has been previously published in Zúñiga et al. (2016), the authors have decided to show them because they are completely necessary to discern between downwelling and upwelling main oceanographic features.

Technical corrections
P. 6, l. 2, bracket missing.

**Response:** Done

---

## Editor Comment (EC1) · L.J. de Nooijer (Editor) · 6 Oct 2016

Dear authors and reviewers,

I apologize for the delay in uploading my comments. After the response of the authors to the reviews, I expected to see an updated version of the manuscript, which can apparently only be asked for if the editor has invited the authors to do so. Anyway, I have read the comments of the reviewers, which I hope the authors have used to improve their manuscript in the way their response suggests. I would like to stress that I agree with reviewer #2, that the authors have to show clearly that the dataset they have, despite its occasional poor temporal coverage, does lead to a (clear) seasonal signal in the production of benthic versus pelagic diatoms. This is one of the crucial

points for the manuscript to be acceptable for publication in Biogeosciences. Perhaps equally important, is my own dissatisfaction about the paleoceanographic relevance of this study. The authors end the discussion with the supposed importance of these findings in reconstructing past climates/ environments. To clarify the potential impact of this study, the authors need to make it more clear how their data/ results can directly improve diatom-based reconstructions. I am looking forward to see a revised version of the manuscript, which I will ask to be reviewed by the original two reviewers, and when necessary, a third reviewer.

Sincerely,

Lennart de Nooijer

---

## Author Comment (AC3) · 16 Oct 2016

Dear Editor:

Please find enclosed the revised version of the manuscript. We think all comments by the reviewers have been successfully justified.

We really appreciate both reviewers and editor suggestions. They really improved the new version of the manuscript.

Sincerely,

Dra. Zúñiga

[Figure]

Please also note the supplement to this comment:
http://www.biogeosciences-discuss.net/bg-2016-201/bg-2016-201-AC3-
supplement.pdf

**Supplement:**

**Diatoms as a paleoproductivity proxy in the NW Iberian coastal upwelling system (NE Atlantic)**

Diana Zúñiga1,2, Celia Santos3,4, María Froján2, Emilia Salgueiro3,5, Marta M. Rufino3,5, Francisco De la Granda6. Francisco G. Figueiras2, Carmen G. Castro2, Fátima Abrantes3,5

5

15

1 University of Vigo, Applied Physics Department, Campus Lagoas Marcosende, E-36310, Vigo, Spain
 2 Consejo Superior de Investigaciones Científicas (CSIC), Instituto de Investigaciones Marinas (IIM), E-36208, Vigo, Spain
 3 Instituto Português do Mar e da Atmosfera (IPMA), Div. Geologia e Georecursos Marinhos, 1495-006, Lisbon, Portugal
 4 MARUM, Center for Marine Environmental Sciences, University of Bremen, 28359, Bremen, Germany.

5 CCMAR - Centre of Marine Sciences. Universidade do Algarve, Campus de Gambelas, 8005-139 Faro 6 Bundesamt für Seeschifffahrt und Hydrographie, Bernhard-Nocht-Str. 78, 20359, Hamburg, Germany

Correspondence to: Diana Zúñiga (diana.zuniga@uvigo.es; imissons@gmail.com)

**Abstract.** The objective of the current work is to improve our understanding as to how diatoms species determine primary production signal in exported and buried particles. We evaluated how the diatom's abundance and assemblage composition

- is transferred from the photic zone to seafloor sediments. To address this, we used a combined analysis of water column, sediment trap and surface sediment samples recovered in the NW Iberian coastal upwelling system was used. Diatom fluxes  $(2.2 \pm 5.6 \ 10^6 \ valves \ m^{-2} \ d^{-1})$  represented the majority of the siliceous microorganisms sinking out from the
- Diatom nuxes (2.2 ± 5.0 10 varies in 'd') represented the majority of the sinceous incroorganisms sinking out from the photic zone during all studied years and showed strong seasonal variability. Discrepancies between water column-sediment trap diatom abundances were found, as shown by the unexpectedly high diatom export signals registered during low productive downwelling periods. They were principally related to surface sediment remobilization and intense Minho and Douro riverine discharges that constitute an additional source of particulate material to the inner continental shelf. Contributions of allochthonous particles to the sinking material were confirmed by the significant increase of both benthic and freshwater diatoms in the sediment trap assemblage.
- 25 Nevertheless, during highly productive upwelling periods no water column-sediment trap direct correlations were found in terms of absolute numbers. However, diatom species sinking out from the photic zone (principally represented by *Chaetoceros* and *Leptocylindrus* spp. resting spores) agreed with those species dominating the water column. This demonstrates that the prevalence of these highly resistant resting spores in the sediments reflect the dominance of both diatom taxa in the surface photic layer when primary production is seasonally intensified. Moreover, our data shows that
- 30 Chaetoceros spp. resting spores dominated the sediment trap assemblage under persistent upwelling winds, high irradiance levels and cold and nutrient-rich waters, while Leptocylindrus spp. spore fluxes were favoured when northerly winds relaxed, and surface water warming promoted water column stratification. Further, this finding will enable the use of relative abundance of both groups in the sediment records as a proxy of persistent vs. intermittent upwelling conditions, which is of particular relevance to infer climatic and oceanographic conditions in the past.

Keywords: diatoms; coastal upwelling; organic carbon; biogenic silica; sediment trap; NW Iberian;

**1** Introduction**

The ocean plays a critical role in the global carbon cycle as a vast reservoir that takes up a substantial portion of the anthropogenically-released carbon from the atmosphere (LeQuéré et al., 2009). A key aspect to understand the ocean carbon

5 cycle includes the role of diatoms as sinkers of primary-produced organic carbon and biogenic silica from the surface productive layer to the sediment record (Sancetta, 1989; Smetacek, 1999; Boyd and Trull, 2007; Romero and Armand, 2010; Tréguer and De La Rocha, 2013). This point underpins the importance and effectiveness of diatom species as productivity indicator in Earth's climate system studies.

Despite the significant advances in this topic, primary production paleoreconstructions via diatom species still require regional calibrations to better understand their response to particular environmental conditions, and to analyse which species transfer the primary production signal via exported and buried particles. Coastal upwelling systems, as sites of major primary production with a marked seasonality are thus ideal for these types of studies (Walsh, 1991; Falkowski et al., 1998; Capone and Hutchins, 2013). The importance and the effectiveness of diatoms as productivity indicators at longer time scales in highly productive coastal regions have been shown by many long-term continuous datasets (Sancetta, 1995; Lange et al., 1998; Romero et al., 2002; Abrantes et al., 2002; Venrick et al., 2003; Onodera et al., 2005).

- In the NW Iberian margin where seasonal upwelling favouring winds generate high primary production rates through modulation of the microplankton community structure (Figueiras and Pazos, 1991; Nogueira and Figueiras, 2005; Espinoza-González et al., 2012), several works have assessed the diatom species ecology in terms of its environmental conditions by comparing the recent sediment record to the hydrographic conditions (Abrantes, 1988; Bao et al., 1997; Abrantes and Moita,
- 20 1999; Gil et al., 2007; Bernárdez et al., 2008; Abrantes et al., 2011). Those authors concluded that the spatial distribution of the sedimentary diatom abundance and assemblages' composition reflects the hydrographic upwelling patterns and primary production trends, with *Chaetoceros* resting spores appearing as a good tracer of the upwelling regime. The aim of this study is to understand the seasonal mechanisms regulating diatom production and export from the photic

zone into the seafloor sediments in the NW Iberian coastal upwelling system, through the combined analysis of diatom 25 abundances and assemblages composition in the water column, sediment trap and surface sediment samples.

**2 Regional setting**

[revised manuscript text omitted]

- 20 maximum levels and the predominant genera in the water column alternated between *Chaetoceros* and *Leptocylindrus* spp. Other species frequently associated with upwelling favourable conditions (e.g. *Asterionellopsis glacialis, Detonula pumila* or *Guinardia delicatula*), appeared sporadically and with lower abundances (Fig. 4 and Table 3).

**4.2 Sinking particulate material time series**

The biogenic silica flux time series, as registered by the trap, contributes from 2% to 10% of the total material, and is closely follow by the siliceous organism fluxes calculated from microscopic counting (Fig. 5a, 5b and 5c). The contribution of diatoms to total siliceous microorganisms dominated throughout the entire period (Fig. 5c and 5e). Only during the 2012 upwelling season did silicoflagellates become relevant, achieving a relative abundance > 7 % (Fig. 5d). Maximum total diatom fluxes were registered under downwelling conditions (Fig. 5e). During these periods benthic and freshwater diatoms became relevant, contributing to the total diatom fluxes up of 24 % and 17 %, respectively (Fig. 6). On the contrary, during upwelling phases total diatom fluxes were relatively low, ranging around a mean seasonal value of  $6 \pm 10 \ 10^5$  valves m-2 d-1 (Fig. 5e and Table 3). During these periods, the diatom assemblage found in the trap samples were mainly composed of *Chaetoceros* spp. and *Leptocylindrus* spp. resting spores, with mean contributions to total marine diatom fluxes of 46 % and 20 %, respectively (Fig. 7 and Table 3).

4.3. Surface sediment samples

5

Diatom abundances in GeoB 11002-1 top sediment sample was  $142 \times 10^4$  valves g-1. Marine diatom assemblage was dominated by resting spores of both *Chaetoceros* (33%) and *Leptocylindrus* (37%) spp., and *Paralia sulcata* (17%) (Table 3). Benthic and freshwater diatoms had contributions < 4 %.

**10 4.4 Relationships between sediment trap main diatom groups and environmental variables**

Canonical correspondence analysis (CCA) stepwise procedure identified five significant variables for the abundance of the main diatom groups (p-value < 0.05), Minho River flow (Minho River), temperature (Temp), Chlorophyll *a* (Chl *a*), NO3 and Si(OH)4 (Fig. 8). The first two canonical axes explained 48.7 % and 40.4 %, i.e. 89 % of the modelled inertia and consequently only those two axes were considered. The CCA model with the five variables explained 46% of the total

- 15 inertia. The first canonical axis showed a positive gradient with Temp and Chl *a* opposite to Minho River discharge. Freshwater (FW) diatoms, benthic diatoms and *Paralia sulcata* (Parsul) were negatively positioned in the first canonical axis, indicating thus a positive relationship with the Minho River, and a negative relationship with temperature and Chl *a*. The second canonical axis showed a negative gradient with NO3 and Si(OH)4 and a negative relationship between these variables and *Chaetoceros* resting spores (ChaeRS). Conversely, *Leptocylindrus* resting spores (LepRS) were positively
- 20 related with NO3 and Si(OH)4. The temporal distribution of the sediment trap samples confirmed that FW diatoms, benthic diatoms and Parsul occurred mainly during downwelling months while ChaeRS and LepRS were associated to upwelling periods (Fig. 8). In addition, this figure also identifies LepRS with late summer and IPC periods.

**5** Discussion**

The siliceous microorganism fluxes determined from the sediment trap, mostly represented by diatoms, showed significant discrepancies with water column diatom abundances (Fig. 3c, 5c and 5e). Such discrepancies were determined by the NW Iberian inner continental margin hydrodynamics, as explained in detail by Zúñiga et al. (2016). These authors described how maximum particle fluxes occurring during downwelling periods were associated with allochthonous sources, explaining the a priori contradictory observation of maximum diatom fluxes during autumn-winter, when irradiance conditions were unfavourable for phytoplankton growth, and Chl *a* showed minimum levels (Fig. 2, 3c and 5e). Furthermore, these authors also showed how seasonal intensification of primary production promoted biogenic settling particles during spring-summer seasons, clarifying why diatom assemblages' dominant species recorded in the trap material were the same as in the water

5 column (Fig. 4 and 7). With this in mind, our results confirm the major influence of both hydrodynamic and biogenic processes over the diatoms abundance, assemblage composition and export in this coastal upwelling system. The implications of these aspects over the use of diatoms as a proxy of paleoproductivity are discussed hereafter.

**5.1. Sediment trap diatom assemblage as a tracer for allocthonous sources in sinking material**

Maximum fluxes of benthic diatoms, whose natural habitat is the sediment interface, run parallel with higher wave heights during highly hydrodynamic downwelling periods (Fig. 2c and 6a). This finding along with the fact that during these highenergy episodes lithogenic particle fluxes achieved their maximum levels (as shown in Zúñiga et al., 2016) confirms that strong storms resuspended surface sediments covering the Iberian continental shelf (Dias et al., 2002; Vitorino et al., 2002. Jounneau et al., 2002, Oberle et al., 2014). Furthermore, stormy conditions were accompanied by intense Minho and Douro River discharges which had a significant effect over the water column thermohaline structure (Fig. 2c and 2d). The

15 significant increase in freshwater diatoms associated to river runoff also confirms continental inputs as an additional source of terrestrial material to the inner continental shelf (Fig. 2d and 6c). Indeed, canonical analysis of sediment trap samples revealed a high correlation between benthic and freshwater diatoms, corroborating the co-ocurrence of both resuspension processes and river discharges during downwelling periods (Fig. 8 and Table 2).

One additional evidence of resuspension resulted from the analysis of marine diatom assemblage collected in the sediment

- 20 trap. *Paralia sulcata* was sporadically found in the water column diatom assemblage during the 2009-2012 studied years (Fig. 4c and Table 3). This meroplanktonic and shadow species, was by contrast, common in ediments and contributed significantly to the trap diatom fluxes during downwelling phases (Fig. 7c and Table 3). As pointed out by previously published sediment trap data from the adjacent Ría de Vigo, this species can be easily resuspended from the sediments under highly hydrodynamic conditions (Bernárdez et al., 2010; Zúñiga et al., 2011). This is, in fact, also supported by the positive
- 25 relationship found between *Paralia sulcata* and benthic diatoms in the trap samples (Fig. 8 and Table 2). Also of interest is the positive correlation between freshwater and benthic diatoms to *Thalassiosira eccentrica* (Table 2), a species which is known to occur in areas where nutrient input is continuous throughout the year, such as in areas influenced by river discharge (Moita, 1993, Abrantes and Moita, 1999).

**5.2 Seasonal succession of diatom species during upwelling seasons: the imprint of the fossil diatom assemblage**

During the studied period, the living diatom community was strongly linked to seasonality revealed by environmental variables, with the highest abundances always recorded during upwelling favourable periods, when irradiance and water column characteristics promote favourable conditions for diatom growth (Fig. 2 and 3).

- 5 A detailed analysis of the marine diatom assemblage during upwelling productive seasons revealed that most living species linked to upwelling favourable conditions were either not present (e.g. *Asterionellopsis glaciallis, Detonula pumila, Guinardia delicatula* and *Skeletonema costatum*) or appeared with a significantly lower contribution in the diatom assemblages (e.g. *Nitzschia* spp., *Pseudo-nitzschia* spp. and small centric) in both sediment trap and the surface sediment sample (Table 3). This observation points to selective dissolution processes acting on thin-walled, less silicified diatoms. As
- 10 a result, the robust and heavily silicified frustules, not only have a ballast effect that promotes a faster arrival to the sediments, but they also have a higher preservation potential in seafloor sediments (Alexander, 1990; Raven and Waite, 2004). Indeed, diatoms assemblages in both sediment trap and surface sediment samples were mainly composed of highly dissolution resistant *Chaetoceros* and *Leptocylindrus* spp. resting spores (Table 3). This fact confirms that these diatom genera are a good sedimentary indicator of highly productive upwelling conditions. Indeed, the sink of *Chaetoceros* and
- 15 Leptocylindrus spp. resting spores, occurring in close correlation with the dominance of both diatom groups in the water column assemblage during the upwelling periods brings new important information to previous works carried out along the Iberian margin, which have only considered *Chaetoceros* spp. resting spores group as tracer of the coastal upwelling regime (Abrantes, 1988; Bao et al., 1997; Abrantes and Moita, 1999).

Canonical analysis performed for the sediment trap data confirms the relationship between the relative abundances of both

- 20 Chaetoceros and Leptocylindrus spp. resting spores with upwelling favourable conditions (positively positioned in CCA1) (Fig. 8). However, they occurred at different times as previously described by other works based on water column data (Figueiras and Rios, 1993; Escaravage et al., 1999; Casas et al., 1999; Nogueira and Figueiras, 2005). The sink of *Chaetoceros* spp. resting spores into the sediment trap was mostly associated to the onset of the upwelling period when irradiance conditions are favourable and persistent northerly winds lead to the upwelling of nutrient-rich subsurface
- ENACW waters on the shelf. On the contrary, *Leptocylindrus* spp. resting spores fluxes were significantly associated to late-summer autumn when more frequent relaxation of winds promoted water column stratification and nutrient depletion (Fig. 8).

In summary, even though the remobilization of bottom sediments by resuspension processes at the inner NW Iberian continental shelf does not allow for a quantitative evaluation of the water column/sediment record preservation efficiencies,

[revised manuscript text omitted]

|                                                        |            | Nitzs.    | Pseudo-     | Thal.              | Small      | Nav.           | Paralia     | Chaeto.              | Lepto.      | Aster.         | Deton.  | Guin.         | Skelet.      |
|--------------------------------------------------------|------------|-----------|-------------|--------------------|------------|----------------|-------------|----------------------|-------------|----------------|---------|---------------|--------------|
|                                                        | Total      | spp.      | nitzs. spp. | nitzs.             | centric    | spp.           | sulcata     | spp                  | spp         | glac.          | pum.    | del.          | cost.        |
| WATER column                                           |            |           |             |                    |            |                |             |                      |             |                |         |               |              |
| Upwelling                                              |            | 1.5 (50)  |             |                    |            |                |             |                      |             |                |         |               |              |
| 1T-1 (1 GD)                                            | 717 (19(0) | 17 (50)   | 25 (20)     | 10 (17)            | 496 (1766) | 1 (2)          | 0.(0)       | 72 (108)             | 12( (271)   | 2 (0)          | 17 (15) | 11            | 1 (2)        |
| $(\pm SD)$                                             | /1/(1809)  | 3(4)      | 25 (30)     | 10(17)
0.1(0.4) | 480 (1700) | 1(2)
01(04) | 0(0)        | 72 (108)             | 26 (33)     | 5 (8)
1 (1) | 3 (9)   | (12)
2 (4) | 1(2)
2(9) |
| Downwelling                                            |            | 5(4)      | 10 (24)     | 0.1 (0.4)          | 20 (51)    | 0.1 (0.4)      | 0(0)        | 27 (20)              | 20 (55)     | 1(1)           | 5())    | 2 (4)         | 2())         |
| cel mL -1 ( $\pm$ SD)                       | 34 (49)    | 2(2)      | 12 (21)     | 1(1)               | 10(18)     | 0(0)           | 1(1)        | 1(1)                 | 1(1)        | 1(1)           | 1(0)    | 0(0)          | 0(0)         |
| % (± SD)                                               | . /        | 12 (12)   | 18 (28)     | 2 (4)              | 52 (25)    | 2 (4)          | 2 (4)       | 7 (10)               | 1 (3)       | 2 (3)          | 2 (7)   | 0 (0)         | 0 (0)        |
| TRAP                                                   |            |           |             |                    |            |                |             |                      |             |                |         |               |              |
| Upwelling                                              |            |           |             |                    |            |                |             |                      |             |                |         |               |              |
| 10 4 valves m -2 d -1 | 50 (05)    | 0.2 (0.1) |             |                    |            |                | 5 0 (I D)   | 10 0 (0 7 0 ) |             | 0 (0)          |         | 0 (0)         | 0 (0)        |
| $(\pm SD)$                                             | 59 (97)    | 2(1)      | 1.1 (5.8)   | 0.7 (1.6)          | 0.5 (0.7)  | 0.1 (0.3)      | 5.3 (13)    | 40.3 (87.2)          | 23.0 (49.9) | 0(0)           | 0(0)    | 0(0)          | 0(0)         |
| % (± SD)
Downwolling                                |            | 2(1)      | 0(0)        | 2 (4)              | 2(2)       | 1(1)           | 14 (15)     | 46 (25)              | 20 (22)     | 0(0)           | 0(0)    | 0(0)          | 0(0)         |
| $10^4$ valves m -2 d -1          |            | 0.1(0.2)  |             |                    |            |                |             |                      |             |                |         |               |              |
| (± SD)                                                 | 216 (462)  | 0.1 (0.2) | 0.1 (0.2)   | 0.9(2.1)           | 1.3 (2.1)  | 0.3 (0.5)      | 31.2 (43.6) | 25.6 (40.6)          | 43.9 (111)  | 0 (0)          | 0(0)    | 0(0)          | 0(0)         |
| % (± SD)                                               |            | 1(1)      | 1 (1)       | 2 (2)              | 3 (2)      | 1(1)           | 23 (12)     | 28 (19)              | 24 (19)     | 0 (0)          | 0 (0)   | 0 (0)         | 0 (0)        |
| SEDIMENT                                               |            |           |             |                    |            |                |             |                      |             |                |         |               |              |
| GeoB 11002-1                                           |            |           |             |                    |            |                |             |                      |             |                |         |               |              |
| 10 4 valves g -1                 | 142        | 0         | 0           | 2                  | 3          | 0              | 24          | 47                   | 52          | 0              | 0       | 0             | 0            |
| %                                                      |            | 0.2       | 0           | 2                  | 2          | 0              | 17          | 33                   | 37          | 0              | 0       | 0             | 0            |

**Figure captions**

Fig. 1. Map of the NW Iberian Peninsula continental margin showing the position of the mooring line (RAIA) site. WANA hindcast reanalysis points 3027034 (WANAS off Cape Silleiro) and 1044067 (WANAG off A Guarda) from which wave data were obtained, location of the irradiance Cies station (IR) and position of the core-top sediment sample GeoB11002-1 are

5 also shown.

[revised manuscript text omitted]

Figure 1

---

## Editor Comment (EC2) · L.J. de Nooijer (Editor) · 26 Oct 2016

Review of BG-2016-201

Dear Dr Zuñiga and co-authors,

I have read the updated version of your manuscript. It has improved considerably, although after some close reading of mine, my opinion remains that you need to make some more thorough changes to the manuscript before acceptance in Biogeosciences. First of all, in my previous assessment, I asked you to use your results to improve paleoceanographic reconstructions using diatoms. One of the reasons for asking this, is that approximately half of the introduction focusses on the use of diatoms for paleoceanographic reconstructions. Could paleosamples be included as 'additional' samples to the CCA (figure 8) to determine what past conditions were (given those included here in the analysis). What would roughly be the uncertainty related to such an approach and what could be done to further improve the applicability of diatoms as reconstruction tools?

Second, and as stated earlier by one of the reviewers, the results presented here should be more directly be compared to the results of previous studies reporting (long-term) monitoring studies. Issues that need to be discussed include: are total fluxes comparable to those of other studies? What are the (dis)similarities between the relative abundances reported here and those of other studies? The CCA shows the correlation between some species and some environmental parameters: is this also reported in other studies? And if there are (large) discrepancies, what could have caused them? Are there environmental parameters that were not included in the analysis that are known from other studies to have a large influence on diatom distribution?

Below, I have added some more, minor comments that may help to improve the manuscript.

Abstract

The first sentence of the abstract is a bit confusing: it is difficult to see how "diatom species could determine the primary production signal...". I think this reads better as something like: "...how the community composition of diatoms reflects sea surface conditions..." Or something similar. This would also make the second sentence of the abstract redundant. Line 17: remove "was used" Line 18 and throughout the text: "2.2 $\pm$5.6 106" is a bit confusing. "2.2 ($\pm$5.6) * 106" would be more clear. Line 19: remove "strong" Line 19-20: discrepancies usually refer to unexpected/ unusual differences between multiple items. I guess here the authors imply discrepancies between different sediment trap samples, although that may be better described as "variability". Or do the authors imply that there is a real discrepancy between the totality of the sediment

trap samples and another dataset? Line 25-26: it is unclear what is supposed to correlate... Absolute numbers are not correlated to what exactly? Lines 25-30: it should become clear that sediment trap-data were compared to diatoms retrieved from core-top material. Line 32: the use of "Further" is inappropriate here. Line 33: please write "vs." in full.

Introduction Line 5-8: the end of this paragraph suggests that this study will somehow deal with the global contribution of diatoms in exporting carbon and Si to the seafloor, which it doesn't. These sentences should reflect the overall aim of this study and should connect to the main conclusions. It also doesn't link to the first sentence of the second paragraph. Line 9-15: it is not clear from the text why there should be a need for regional calibrations. Probably best to rephrase this paragraph: there are numerous long-term studies, which have shown that there are considerable differences between regions. Then, why is it particularly interesting/ necessary to study the Iberian Margin? Are regions with clear seasonal upwelling not covered in the listed long-term studies? Line 20-22: so, if other authors already showed that diatoms from core-top samples reflect those that are found in the overlying water column, what is the need of this particular study?

Material and methods

External forcing Line 13-14: replace "accessed via" by "available through". Line 25: not all readers may be familiar with "Puertos del Estado". Please explain what this is.

Water column Line 27: replace "on board" by "by". Line 29: assuming that the Niskin bottles are made of PVC and have a volume of 10 liters, please put a space after the "L" in "10-LPVC". Line 7: from what depths were the samples taken for determining the diatom abundances? Were these depths sampled every single time? Were the samples combined before analysis of the diatom species assemblage? If not, did the authors find consistent differences between water depths? Line 13-14: this sentence is redundant: it also appears at the end of section 3.3

Surface sediments Line 13-17: how was the sample taken? To what depth was the box core sampled? Do the authors have an idea about the sedimentation rate in this area and thereby, have an idea about the age that the diatoms may cover? Was there only one sample taken? If this is the case, could the authors make clear why there is no influence of spatial variability?

Statistical data analysis Line 19: please remove the second "between".

Results

Environmental conditions Line 8-22: please add a description of the variability between years. I.e. are the observed trends consistent between years? Line 15-16: what exactly is the uncertainty here? Is this the standard deviation? If this is the case, the variability between samples must be very high and there should be a report here of minimum and maximum values in addition to the average values. Line 16-17: what do "exceptionally" and "relevant" mean here? Line 18: should be "lead" instead of "leaded"

Sinking particulate material Line 24-5: the description does not mention (variability in) absolute numbers as found in the samples, only the relative numbers. A brief description of the trends in absolute numbers should also be included. Line 25: should be "followed".

Relationships between sediment trap main diatom groups Line 11-22: why are the samples from the water column not added to the CCA?

Discussion

Sediment trap diatom assemblage Line 19: "One additional evidence" is not correct English. Please rephrase. Line 9-28: this section lacks a thorough comparison to previous (long-term) monitoring studies on diatom assemblages, which needs to be included in the discussion.

Seasonal succession of diatom species As stated before, this section (or an entire new one) needs to make clear what this dataset can add to the use of diatom assemblages as reconstruction tools. With the statistical analysis presented in this section, the authors should be able to propose a (quantitative) use of such assemblages to reconstruct upwelling/ downwelling conditions.

Figures The lighter two colors are difficult to distinguish in figure 4. Figure 8 can be improved too by enhancing the contrast in the symbols used. The captions of figures 3-7 should explicitly state whether the figure displays CTD- or sediment trap samples.

---

## Author Response (AR1)

**Review of BG-2016-201**

Dear Dr Zúñiga and co-authors,
I have read the updated version of your manuscript. It has improved considerably, although after some close reading of mine, my opinion remains that you need to make some more thorough changes to the manuscript before acceptance in Biogeosciences. First of all, in my previous assessment, I asked you to use your results to improve paleoceanographic reconstructions using diatoms. One of the reasons for asking this, is that approximately half of the introduction focusses on the use of diatoms for paleoceanographic reconstructions. Could paleosamples be included as 'additional' samples to the CCA (figure 8) to determine what past conditions were (given those included here in the analysis). What would roughly be the uncertainty related to such an approach and what could be done to further improve the applicability of diatoms as reconstruction tools? Second, and as stated earlier by one of the reviewers, the results presented here should be more directly be compared to the results of previous studies reporting (longterm) monitoring studies. Issues that need to be discussed include: are total fluxes comparable to those of other studies? What are the (dis)similarities between the relative abundances reported here and those of other studies? The CCA shows the correlation between some species and some environmental parameters: is this also reported in other studies? And if there are (large) discrepancies, what could have caused them? Are there environmental parameters that were not included in the analysis that are known from other studies to have a large influence on diatom distribution?

First of all, the authors would like to gratefully thank all comments and suggestions from the editor. They undoubtedly improve the quality of this manuscript.

In agreement with first editor´s comment, the authors consider that it would be very interesting to include paleosamples in the CCA. However, this is conceptually and methodologically not possible. CCA was carried out between the diatom assemblages from the sediment trap samples (dependent variables) and the environmental factors (independent variables) with the aim to evaluate how the environmental conditions were related with diatoms export from the surface productive layer. To do that they used interpolated water column environmental data for the time interval recovered by each trap sample (see section 3.5 for further details). It is obvious that such environmental data are not available for the paleosamples. Indeed, our sediment trap-environmental data calibration will bring the opportunity to infer paleoenvironmental conditions over geological timescales from using downcore fossil diatom assemblages. This is the goal of this work and why the authors consider it so relevant for the scientific community working in the NW Iberian margin, the most important coastal upwelling region of Europe.

On the other hand, answering the question raised by the editor with regards the comparison with other studies carried out in other coastal upwelling systems, the authors would like to underline that even though values were similar in terms of "orders of magnitude", they are not really comparable. Long-term diatom flux data in other coastal upwelling systems were recovered at locations further offshore and deeper, complicating a direct comparison with data presented here. For that reason, the authors would propose just to reference previous studies in the introduction section. However, whether editor considers such comparison absolutely necessary the authors would propose to initiate discussion section with a sentence like that:

*Diatoms exported out from the photic zone (2.2 (±5.6) 106 valves m-2 d-1), similar to those registered in other coastal upwelling systems (Sancetta, 1995; Lange et al., 1998; Romero et al., 2002; Abrantes et al., 2002; Venrick et al., 2003; Onodera et al., 2005) showed contrasting results compared to diatom abundances on the surface waters (Fig. 3c, 5c and 5e).*

The authors would also like to state that to their knowledge there are no studies that made a CCA to correlate diatom fluxes and environmental variables. Such statistical analysis is frequently used on studies made over water column samples, but not over sediment trap samples. Indeed, this probe the value and worth of the data presented here, since it demonstrates for the first time how the relationship between water column diatoms assemblage and environment is also transferred to diatoms exported from the photic layer. In this regard, the authors would also like to highlight that they consider environmental variables dataset used in this study a fairly complete picture of factors affecting diatoms development and export in the marine environment. They want to remember that even only significant variables were presented in the CCA, a complete dataset of environmental parameters were considered (see section 3.5).

The authors encourage editor to read the new version of the manuscript where all suggestions pointed out by him/her were taken into account. They really consider that this new version really highlights the goal of this work. The abstract and both the introduction and discussion sections were modified in order to achieve this. In addition, main conclusions derived from this work were summarized in this new version.

Below, I have added some more, minor comments that may help to improve the manuscript.

Abstract
The first sentence of the abstract is a bit confusing: it is difficult to see how "diatom species could determine the primary production signal...". I think this reads better as something like: "...how the community composition of diatoms reflects sea surface conditions..." Or something similar. This would also make the second sentence of the abstract redundant.
The authors agree with editor´s comment. First sentence has been modified in the new version of the manuscript.

Line 17: remove "was used"
Done

Line 18 and throughout the text: "2.2 ± 5.6 106" is a bit confusing. "2.2 (±5.6) * 106" would be more clear.
Done

Line 19: remove "strong"
Done

Line 19-20: discrepancies usually refer to unexpected/ unusual differences between multiple items. I guess here the authors imply discrepancies between different sediment trap samples, although that may be better described as "variability". Or do the authors imply that there is a real discrepancy between the totality of the sediment trap samples and another dataset?

The authors would like to clarify that the term "discrepancies" is not referred to variations between sediment trap samples nor to the comparison with another dataset, but indicates unexpected results when comparing water column and sediment trap diatom´s abundances seasonally. In any case, this term has been modified by "contrasting results".

Line 25-26: it is unclear what is supposed to correlate... Absolute numbers are not correlated to what exactly?
With this sentence the authors wanted to highlight that diatom fluxes registered by the sediment trap moored at the base of the photic zone did not reflect total diatoms abundance in the surface layer. In any case, sentence has been modified to make this statement clearer.

Lines 25-30: it should become clear that sediment trap-data were compared to diatoms retrieved from core-top material.
The authors totally agree with editor´s comment. Last paragraph of the abstract has been modified to remark this important aspect.

Line 32: the use of "Further" is inappropriate here.
This term has been eliminated in the new version of the manuscript

Line 33: please write "vs." in full.
Done

Introduction
Line 5-8: the end of this paragraph suggests that this study will somehow deal with the global contribution of diatoms in exporting carbon and Si to the seafloor, which it doesn't. These sentences should reflect the overall aim of this study and should connect to the main conclusions. It also doesn't link to the first sentence of the second paragraph.
Attending to editor suggestion this paragraph has been modified. The authors consider new paragraph better reflects the overall aim of our study, and otherwise also connect to the main conclusions.

Line 9-15: it is not clear from the text why there should be a need for regional calibrations. Probably best to rephrase this paragraph: there are numerous long-term studies, which have shown that there are considerable differences between regions. Then, why is it particularly interesting/ necessary to study the Iberian Margin? Are regions with clear seasonal upwelling not covered in the listed long-term studies?
This paragraph has been modified attending to the issues raised.

Line 20-22: so, if other authors already showed that diatoms from core-top samples reflect those that are found in the overlying water column, what is the need of this particular study?
The need of this study is justified since studies based on core-top samples achieved their conclusions based on discrete water column data recovered during isolate cruises that directly were compared with surface sediment samples spatially distributed along the margin. So, from author´s perspective, they consider these studies "unfinished" because of the limitation of water column data to infer diatoms export out from the surface productive layer in seasonal terms. In that sense, their multi-year sediment trap dataset allowed the link between the seasonal successions of living diatoms community with the fossil

diatom assemblage registered on the seafloor sediments. With their data the authors not only show that *Chaetoceros* and *Leptocylindrus* spp. resting spores in the sediments marks upwelling favourable conditions but expose that each diatom genera reflect different environmental conditions in response to different phases of the upwelling regime. This is the goal of this work. The authors have modified abstract and discussion in order to clearly submit this message to the audience.

Material and methods
External forcing
Line 13-14: replace "accessed via" by "available through".
Done

Line 25: not all readers may be familiar with "Puertos del Estado". Please explain what this is. Water column
Web page has been included in the new version of the manuscript in order to provide the reader with additional information.

Line 27: replace "on board" by "by".
Done

Line 29: assuming that the Niskin bottles are made of PVC and have a volume of 10 liters, please put a space after the "L" in "10-LPVC".
Done

Line 7: from what depths were the samples taken for determining the diatom abundances? Were these depths sampled every single time? Were the samples combined before analysis of the diatom species assemblage? If not, did the authors find consistent differences between water depths?
As stated in methods section for diatoms counting and identification samples from **5 water depth** were used". This depth was sampled monthly during the experimental years, except during the period January-June 2010. The samples were never combined before the analysis. To determine diatom species assemblage each sample was treated separately as shown in figure 4.

Line 13-14: this sentence is redundant: it also appears at the end of section 3.3. Surface sediments
The authors agree with editor´s comment that the sentence is redundant. However, as suggested by one of the reviewers, they consider it a key aspect because all statistical analysis is based on these criteria.

Line 13-17: how was the sample taken? To what depth was the box core sampled? Do the authors have an idea about the sedimentation rate in this area and thereby, have an idea about the age that the diatoms may cover? Was there only one sample taken? If this is the case, could the authors make clear why there is no influence of spatial variability?
As explained in section 3.4. surface sediment sample was collected with a giant box corer in a station located near RAIA position at 111 m water depth.
In relation with the sedimentation rate for this study area the authors would like to pointed out that this was evaluated from Pb210 analyses carried out over a sediment core located close to our study site

(42.16664, -9.02669; 129 m). From this core, the authors estimated a linear sedimentation rate of 0.0784 cm/yr, meaning a temporal resolution of ca. 13 years in 1cm of sediment. However, they considered these data may be distorted by the strong resuspension processes occurring during autumn-winter downwelling periods. For that reason, they decided not to use these data in the manuscript and just consider the surface sediment sample as representative of the "present".

The authors would also like to clarify that this is not the unique sample recovered along the NW Iberian margin. Indeed, in the first version of the manuscript the authors presented two core top samples across margin. However, as suggested by reviewer 1, only the sample located close to the mooring line should be used.

*Response to reviewer 1 (first version of the manuscript): Higher diatom abundances in the surface sediment at GeoB 11002-1 onshore station responded to primary production signal in the photic layer. Even not presented here the authors have evaluated Chl a contents through a transect perpendicular to the coast during almost the entire sampling period. They observed how seasonal Chl a at the surface productive layer is intensified close to the coast, in agreement with diatom´s abundance in surface sediment samples. This confirms the use of valves/g as a good indicator of diatom´s production in the photic zone. In that sense, since onshore surface sediment sample is closest to the RAIA station the authors consider it better to reflect the conditions at this site and will be the only one used in the new version of the manuscript.*

Statistical data analysis
Line 19: please remove the second "between".
Done

Results
Environmental conditions
Line 8-22: please add a description of the variability between years. I.e. are the observed trends consistent between years?
After a careful reading of the manuscript the authors consider inter-annual description as not necessary. This aspect is not discussed at any time neither in the other result sections nor in the discussion.

Line 15-16: what exactly is the uncertainty here? Is this the standard deviation? If this is the case, the variability between samples must be very high and there should be a report here of minimum and maximum values in addition to the average values.
Done

Line 16-17: what do "exceptionally" and "relevant" mean here?
The term "exceptionally" has been modified to "sporadically" to avoid confusion. Also, the term "relevant" has been replaced by "highly abundant".

Line 18: should be "lead" instead of "leaded" Sinking particulate material
Corrected

Line 24-5: the description does not mention (variability in) absolute numbers as found in the samples, only the relative numbers. A brief description of the trends in absolute numbers should also be included.

Additional information has been included in the new version of the manuscript.

Line 25: should be "followed".
Corrected

Relationships between sediment trap main diatom groups
Line 11-22: why are the samples from the water column not added to the CCA?
The authors have only used sediment trap data for the CCA since they consider strength of this work is to show the link between the environmental and the sink of diatoms as a potential source of microfossils to the seafloor sediments. Indeed, the relationship between the water column diatom community and the environmental variables in this coastal upwelling system has been described in previous works as shown in the discussion.

Nogueira, E., Figueiras, F.G.: The microplankton succession in the Ría de Vigo revisited: species assemblages and the role of weather-induced, hydrodynamic variability. J. Mar. Sys. 54, 139-155, 2005.
Figueiras, F.G., Rios, A.F.: Phytoplankton succession, red tides and the hydrographic regime in the Rias Bajas of Galicia. In: Toxic Phytoplankton Blooms in the Sea. T.J., Smayda and Y. Shimizu, Ed. Elsevier Science Publishers B.V., 239-244, 1993.
Casas, B, Varela, M., Bode, A.: Seasonal succession of phytoplakton species on the coast of A Coruña (Galicia, Northwest Spain). Boletín del Instituto Español de Oceanografía, 15, 413-429, 1999.

Discussion
Sediment trap diatom assemblage
Line 19: "One additional evidence" is not correct English. Please rephrase.
Sentence has been rewritten.

Line 9-28: this section lacks a thorough comparison to previous (long-term) monitoring studies on diatom assemblages, which needs to be included in the discussion.
As explained immediately above long-term diatom flux data in other coastal upwelling systems were recovered at locations further offshore and deeper, complicating a direct comparison with data presented here. For that reason and even values were similar in terms of "orders of magnitude", they would propose not to include these studies in the discussion section, only reference them in the introduction.
However, whether editor considers such comparison absolutely necessary the authors would propose to initiate discussion section with a sentence like that:

Diatoms exported out from the photic zone (2.2 (±5.6) 106 valves m-2 d-1), similar to those registered in other coastal upwelling systems (Sancetta, 1995; Lange et al., 1998; Romero et al., 2002; Abrantes et al., 2002; Venrick et al., 2003; Onodera et al., 2005) showed contrasting results compared to diatom abundances on the surface waters (Fig. 3c, 5c and 5e).

Seasonal succession of diatom species As stated before, this section (or an entire new one) needs to make clear what this dataset can add to the use of diatom assem-blages as reconstruction tools. With the

statistical analysis presented in this section, the authors should be able to propose a (quantitative) use of such assemblages to reconstruct upwelling/ downwelling conditions.

A new section at the end of the manuscript has been added trying to summarize main conclusions of the study. Differences between upwelling/donwelling conditions in relation with diatoms assemblages have been described.

Figures The lighter two colors are difficult to distinguish in figure 4.

Figure has been modified

Figure 8 can be improved too by enhancing the contrast in the symbols used.

Figure has been modified. Maximum contrast has been applied to the symbols. The authors wonder whether it is a colour screen problem.

The captions of figures 3-7 should explicitly state whether the figure displays CTD- or sediment trap samples.

New information has been included in the new version of the manuscript

**Diatoms as a paleoproductivity proxy in the NW Iberian coastal upwelling system (NE Atlantic)**

Diana Zúñiga[1,2], Celia Santos[3,4,5], María Froján[2], Emilia Salgueiro[3,5], Marta M. Rufino[3,5], Francisco De la Granda[6]. Francisco G. Figueiras[2], Carmen G. Castro[2], Fátima Abrantes[3,5]

[1] University of Vigo, Applied Physics Department, Campus Lagoas Marcosende, E-36310, Vigo, Spain
[2] Consejo Superior de Investigaciones Científicas (CSIC), Instituto de Investigaciones Marinas (IIM), E-36208, Vigo, Spain
[3] Instituto Português do Mar e da Atmosfera (IPMA), Div. Geologia e Georecursos Marinhos, 1495-006, Lisbon, Portugal
[4] MARUM, Center for Marine Environmental Sciences, University of Bremen, 28359, Bremen,
[5] CCMAR - Centre of Marine Sciences. Universidade do Algarve, Campus de Gambelas, 8005-139 Faro
[6] Federal Maritime and Hydrographic Agency of Germany, 20359, Hamburg, Germany

*Correspondence to*: Diana Zúñiga (diana.zuniga@uvigo.es; imissons@gmail.com)

**Abstract.** The objective of the current work is to improve our understanding with regards how water column  diatom's abundance and assemblage composition is seasonally transferred from the photic zone to seafloor sediments. To address this, we used a combined analysisset  derived from water column, sediment trap and surface sediment samples recovered in the NW Iberian coastal upwelling system .

Diatom fluxes (2.2 (±-5.6) 10^6 valves m^{-2} d^{-1}) represented the majority of the siliceous microorganisms sinking out from the photic zone during all studied years and showed  seasonal variability.  Contrasting results between water column  -sediment trap diatom abundances were found during downwelling periods, as shown by the unexpectedly high diatom export signals when diatom-derived primary production achieved their minimum levels . They were principally related to surface sediment remobilization and intense Minho and Douro riverine discharges that constitute an additional source of particulate  matter to the inner continental shelf. In fact, contributions of allochthonous particles to the sinking material were confirmed by the significant increase of both benthic and freshwater diatoms in the sediment trap assemblage.

On the other hand, we found that most of the living diatom species blooming during highly productive upwelling periods were dissolved during sinking, and only the resistant to dissolution and the *Chaetoceros* and *Leptocylindrus* spp. resting spores were susceptible to be exported and buried. Furthermore, *Chaetoceros* spp. dominate during spring-early summer, when persistent northerly winds lead to the upwelling of nutrient-rich waters on the shelf, while *Leptocylindrus* spp. appears associated to late summer /upwelling relaxation, characterized by water column stratification and nutrient depletion. These findings evidence that the contributions of these diatom genera to the sediment's total marine diatom assemblage should allow for the reconstruction of different past upwelling regimes.

no water column-sediment trap direct correlations were found in terms of absolute numbers. However, diatom species sinking out from the photic zone (principally represented by *Chaetoceros* and *Leptocylindrus* spp. resting spores) agreed with those species dominating the water column. This demonstrates that the prevalence of these highly resistant resting spores in the sediments reflect the dominance of both diatom taxa in the surface photic layer when primary production is seasonally intensified. Moreover, our data shows that *Chaetoceros* spp. resting spores dominated the sediment trap assemblage under persistent upwelling winds, high irradiance levels and cold and nutrient rich waters, while *Leptocylindrus* spp. spore fluxes were favoured when northerly winds relaxed, and surface water warming promoted water column stratification. Further, this finding will enable the use of relative abundance of both groups in the sediment records as a proxy of persistent vs. intermittent upwelling conditions, which is of particular relevance to infer climatic and oceanographic conditions in the past.

**Keywords:** diatoms; coastal upwelling; organic carbon; biogenic silica; sediment trap; NW Iberian;

**1 Introduction**

Diatoms are the most important primary producers in the ocean and play a key role in biogeochemical cycles through transferring organic carbon and biogenic silica from the surface layer to the seafloor sediments (Sancetta, 1989; Romero and Armand, 2010; Tréguer and De La Rocha, 2013). The preservation of their siliceous valves in marine sediment records has promoted their use as paleoproductivity indicators. However, reconstruction of primary production still suffer from diverse uncertainties, indicating that more studies are needed to accurately discern how particular environmental conditions regulate the diatom response, and how diatom´s ecological traits transfer primary production signal from the water column to the sediments through exported and buried particles. In this regard, the analysis of data provided by sediment traps have contributed significantly to improve our knowledge of this topic, because the deployment of traps still is the best approach for monitoring downward diatom fluxes, as response to oceanographic and biological processes occurring in surface waters on long-term basis.

Coastal upwelling systems are sites with major diatom-derived primary production where seasonality is often a noticeable feature (Walsh, 1991; Falkowski et al., 1998; Capone and Hutchins, 2013). Consequently, many studies focused on how primary production signal is exported through the water column in these highly productive coastal regions were conducted through the analysis of downward diatom flux time series (Sancetta, 1995; Lange et al., 1998; Romero et al., 2002; Abrantes et al., 2002; Venrick et al., 2003; Onodera et al., 2005). In the NW Iberian margin, despite being the most productive upwelling region in Europe (Figueiras and Pazos, 1991; Nogueira and Figueiras, 2005; Espinoza-González et al., 2012), the use of diatoms as a productivity tracer to date was based on a direct comparison of the hydrographic conditions with surface sediment (Abrantes, 1988; Bao et al., 1997; Abrantes and Moita, 1999). From these studies it was concluded that *Chaetoceros* resting spores could be used as a good tracer of upwelling patterns, in particular the position of the upwelling front. Nevertheless, none of these studies provided information on the processes regulating seasonal diatom production and

export from the photic zone to the seafloor sediments. In this context, the aim of this work is to go further in this topic by presenting the first analysis of the diatom community that combines water column, sediment trap and surface sediment samples recorded in this margin. Our results will provide relevant information regarding the use of fossil diatom assemblages as a primary production paleotracer in the highly productive NW Iberian coastal upwelling system.

The ocean plays a critical role in the global carbon cycle as a vast reservoir that takes up a substantial portion of the anthropogenically-released carbon from the atmosphere (LeQuéré et al., 2009). A key aspect to understand the ocean carbon cycle includes the role of diatoms as sinkers of primary produced organic carbon and biogenic silica from the surface productive layer to the sediment record (Sancetta, 1989; Smetacek, 1999; Boyd and Trull, 2007; Romero and Armand, 2010; Tréguer and De La Rocha, 2013). This point underpins the importance and effectiveness of diatom species as productivity indicator in Earth´s climate system studies.

Despite the significant advances in this topic, primary production paleoreconstructions via diatom species still require regional calibrations to better understand their response to particular environmental conditions, and to analyse which species transfer the primary production signal via exported and buried particles. Coastal upwelling systems, as sites of major primary production with a marked seasonality are thus ideal for these types of studies (Walsh, 1991; Falkowski et al., 1998; Capone and Hutchins, 2013). The importance and the effectiveness of diatoms as productivity indicators at longer time scales in highly productive coastal regions have been shown by many long-term continuous datasets (Sancetta, 1995; Lange et al., 1998; Romero et al., 2002; Abrantes et al., 2002; Venrick et al., 2003; Onodera et al., 2005).

In the NW Iberian margin where seasonal upwelling favouring winds generate high primary production rates through modulation of the microplankton community structure (Figueiras and Pazos, 1991; Nogueira and Figueiras, 2005; Espinoza-González et al., 2012), several works have assessed the diatom species ecology – in terms of its environmental conditions by comparing the recent sediment record to the hydrographic conditions (Abrantes, 1988; Bao et al., 1997; Abrantes and Moita, 1999; Gil et al., 2007; Bernárdez et al., 2008; Abrantes et al., 2011). Those authors concluded that the spatial distribution of the sedimentary diatom abundance and assemblages´ composition reflects the hydrographic upwelling patterns and primary production trends, with *Chaetoceros* resting spores appearing as a good tracer of the upwelling regime.

The aim of this study is to understand the seasonal mechanisms regulating diatom production and export from the photic zone into the seafloor sediments in the NW Iberian coastal upwelling system, through the combined analysis of diatom abundances and assemblages composition 
[revised manuscript text omitted]
 (52 ± 25 %) (Fig. 3c, 4b and Table 3). Only exceptionallysporadically, *Navicula* spp. and *Paralia sulcata* become relevantwere highly abundant (Fig. 4) (Fig. 3c, 4

20   and Table 3). On the other hand, from April- May to October, the margin was characterized by high irradiance levels and the upwelling of cold (< 14ºC) and nutrient rich ENACW on the continental shelf, that leadped to the development of Chl $a$ maxima (Fig. 2a, 2b and 3). During theose highly productive upwelling periods, diatom abundances achieved maximum levelswere high (up to 7629 cel mL$^{-1}$) (Figure 3c). Tand the predominant genera 
[revised manuscript text omitted]

---

## Author Response (AR2)

**Editor´s comment**

Dear Dr Zuniga and co-authors,

I have read the revised version of your manuscript, which has improved and is almost fit for publication in Biogeosciences. I was surprised by your statement that it is technically not possible to add paleosamples to the CCA. This is relatively easy to do: any samples for which you have no environmental data, can be plotted in the CCA based on the similarity of the species composition and those of the already plotted samples. The relation between the 'original' sample set and the environmental vectors will not change as these paleosamples are added. This will, however, show the similarity of the paleosamples to the modern ones. Since I think this will be a substantial improvement of your manuscript, I suggest to add modify this figure.

**Authosr´s response to Editor**

Firstly, the authors would like to apologize for the confusion generated with regards this important aspect of the manuscript. They agree with editor that this information is really relevant for data interpretation. They encourage editor to check new version some modifications were included.

As suggested by the editor, surface sediment sample location on the ordination plot has been predicted using CCA model generated from the sediment trap samples. As shown in Figure 8, surface sediment prediction was included in the CCA plot. In that sense, as remarked during the discussion process, the authors would like to clarify that even we are on disposal of more sediment samples along the NW Iberian margin (check first version of the manuscript), they were not included in Figure 8. The reason was that those samples were not considered as representative of our study area for reviewer 1. He/she encourage us to only used the sample located close to the mooring line.

The authors would like underline their gratitude regarding revision procedure. They are totally aware of the editor´s effort during the discussion process.

Please contact us (dianazuniga@iim.csic.es) if you need further information concerning what has been done.

NOTE: As suggested by associated editor, a private email was sent to him last week. On receiving no response the authors have decided to upload documents in Copernicus platform with the time limit set by the journal (08 Feb 2017).

[revised manuscript text omitted]